# Delayed fluorescence from inverted singlet and triplet excited states

Naoya Aizawa[1,2,3✉], Yong-Jin Pu[1,4✉], Yu Harabuchi[5,6], Atsuko Nihonyanagi[1], Ryotaro Ibuka[1], Hiroyuki Inuzuka[1], Barun Dhara[1], Yuki Koyama[1,4], Ken-ichi Nakayama[2], Satoshi Maeda[5,6], Fumito Araoka[1] & Daigo Miyajima[1✉]

Hund's multiplicity rule states that a higher spin state has a lower energy for a given electronic configuration[1]. Rephrasing this rule for molecular excited states predicts a positive energy gap between spin-singlet and spin-triplet excited states, as has been consistent with numerous experimental observations over almost a century. Here we report a fluorescent molecule that disobeys Hund's rule and has a negative singlet–triplet energy gap of $-11 \pm 2$ meV. The energy inversion of the singlet and triplet excited states results in delayed fluorescence with short time constants of 0.2 µs, which anomalously decrease with decreasing temperature owing to the emissive singlet character of the lowest-energy excited state. Organic light-emitting diodes (OLEDs) using this molecule exhibited a fast transient electroluminescence decay with a peak external quantum efficiency of 17%, demonstrating its potential implications for optoelectronic devices, including displays, lighting and lasers.

The spin multiplicity of molecular excited states plays a crucial role in organic optoelectronic devices. In the case of OLEDs, recombination of charge carriers leads to the formation of singlet and triplet excited states in a 1:3 ratio. This spin statistics limits the internal quantum efficiency of OLEDs and leads to the energy loss owing to the spin-forbidden nature of triplet excited states to emit photons. To overcome this issue, two strategies for harvesting the 'dark' triplet excited states as photons have been established. The first relies on organometallic complexes with transition metals, such as iridium and platinum, which induce a large spin–orbit coupling to allow triplet states to emit photons as phosphorescence[2–4]. The other uses organic molecules that exhibit thermally activated delayed fluorescence (TADF)[5–7]. This class of materials has energetically close singlet and triplet excited states, in which ambient thermal energy upconverts the triplet states into the singlet states through reverse intersystem crossing (RISC). Although the concept of TADF has the advantage of eliminating the need for transition metals, the resultant temporally delayed fluorescence typically has a time constant in the microsecond or even millisecond range, which is long enough for detrimental bimolecular annihilations, such as triplet–triplet annihilation and triplet–polaron annihilation, to compete with delayed fluorescence. These bimolecular annihilations lead to the decrease in device efficiency under high current densities, known as efficiency roll-off in OLEDs[8,9], and also generate high-energy excitons that are suspected to cause chemical degradation of materials, particularly in blue OLEDs[10]. The research community has thus focused on minimizing the singlet–triplet energy gap ($\Delta E_{ST}$) to accelerate the upconversion by thermal activation[7]. Alternatively, an ideal case would be thermodynamically favourable downconversion with negative $\Delta E_{ST}$, which is not expected if applying Hund's multiplicity rule to the lowest-energy excited state.

Herein, we demonstrate experimental evidence of the existence of highly fluorescent organic molecules that disobey Hund's rule and possess negative $\Delta E_{ST}$ for constructing efficient OLEDs.

Numerous observations of positive $\Delta E_{ST}$ in molecular excited states are generally understood by the exchange interaction, the quantum-mechanical effect involving Pauli repulsion, which stabilizes triplet states relative to singlet states[11]. $\Delta E_{ST}$ is simply equal to twice the positive exchange energy if the lowest-energy singlet and triplet excited states ($S_1$ and $T_1$) have the same single-excitation configuration[11]. Although there is general agreement that $\Delta E_{ST}$ must be positive, potentially negative $\Delta E_{ST}$ has been discussed in nitrogen-substituted phenalene analogues, such as cycl[3.3.3]azine and heptazine, during the past two decades[12–21]. Recent theoretical studies have also suggested the possibility of negative $\Delta E_{ST}$ in these molecules by accounting for double-excitation configurations in which two electrons of occupied orbitals have been promoted out to virtual orbitals[15–19] (Supplementary Fig. 1). Because the Pauli exclusion principle restricts the accessible double-excitation configurations in $T_1$, an effective admixture of such configurations stabilizes $S_1$ relative to $T_1$. If this stabilization overcomes the exchange energy, $\Delta E_{ST}$ could be a negative value (Fig. 1a). However, to the best of our knowledge, none of the molecules has been experimentally identified with negative $\Delta E_{ST}$ and the resultant delayed fluorescence from inverted singlet and triplet excited states (DFIST). We note that the accounting for double-excitation configurations has proved crucial to theoretically reproduce the small but positive $\Delta E_{ST}$ of 5,9-diphenyl-5,9-diaza-13b-boranaphtho[3,2,1-de]anthracene (DABNA-1) (0.15 eV)[22,23].

Pioneering computational calculations[15] inspired us to focus on heptazine as a potential class of molecules that exhibit DFIST. Correlated wave function theories suggested that $S_1$ of heptazine lies 0.2–0.3 eV

[1]RIKEN Center for Emergent Matter Science (CEMS), Wako, Japan. [2]Department of Applied Chemistry, Graduate School of Engineering, Osaka University, Suita, Japan. [3]Precursory Research for Embryonic Science and Technology (PRESTO), Japan Science and Technology Agency (JST), Kawaguchi, Japan. [4]Graduate School of Organic Materials Science, Yamagata University, Yonezawa, Japan. [5]Department of Chemistry, Faculty of Science, Hokkaido University, Sapporo, Japan. [6]Institute for Chemical Reaction Design and Discovery (WPI-ICReDD), Hokkaido University, Sapporo, Japan. ✉e-mail: aizawa@chem.eng.osaka-u.ac.jp; yongjin.pu@riken.jp; daigo.miyajima@riken.jp

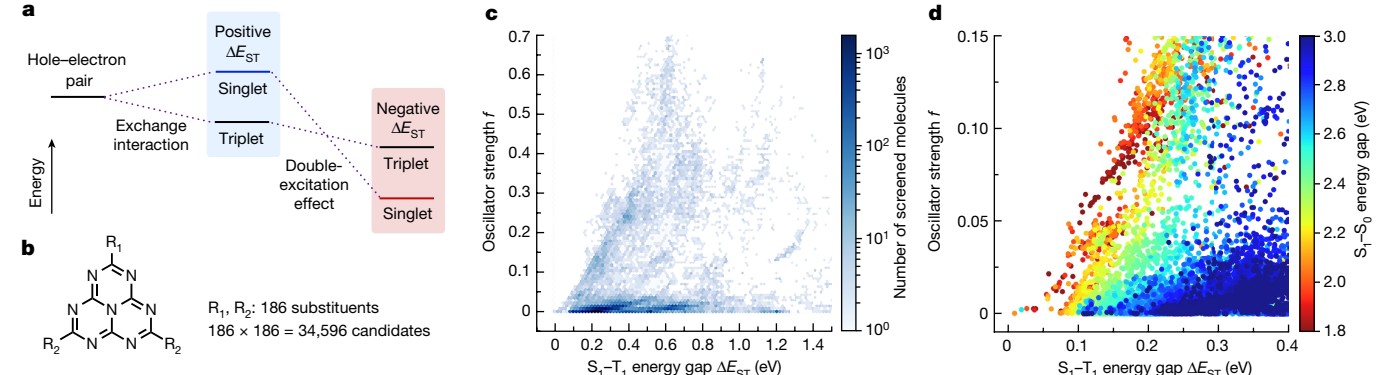

**Fig. 1 | Computational screening of heptazine analogues. a**, Schematic diagram of singlet and triplet excited states split in energy by the exchange interaction (middle) and then inverted by including the double-excitation effect (right). **b**, Molecular structures of the heptazine analogues examined in the computational screening. **c**, Number of screened molecules as a function of $\Delta E_{ST}$ and $f$ calculated by TDDFT. **d**, $S_1$–$S_0$ energy gaps as a function of $\Delta E_{ST}$ and $f$ calculated by TDDFT.

below $T_1$, although $S_1$ is a 'dark' state, meaning that the electronic transition to the ground state ($S_0$) is dipole-forbidden and the oscillator strength ($f$) is zero in the $D_{3h}$ symmetry point group. Notably, the heptazine core is shared by several synthesized molecules that exhibit intense TADF[24,25] with positive $\Delta E_{ST}$ (refs. [26,27]). Furthermore, the recent computational screening by Pollice et al. has demonstrated that appropriate chemical modifications of heptazine recover $f$ while retaining negative $\Delta E_{ST}$ (ref. [19]). As such, we introduced 186 different substituents to heptazine to generate 34,596 candidate molecules for computational screening. The structures of all substituents are available in Supplementary Fig. 2. To ensure the synthetic feasibility, at most two distinct types of substituents were introduced to the heptazine core as $R_1$ and $R_2$ (Fig. 1b). We used standard linear-response time-dependent density functional theory (TDDFT) to calculate $\Delta E_{ST}$ and $f$, which are more affordable in computational cost than those calculated by correlated wave function theories. Although the commonly used adiabatic approximation in TDDFT does not account for double-excitation character[16,28], the properties calculated by TDDFT are still useful to prescreening for narrowing the list of the candidate molecules before the high-cost calculations and experimental evaluation, as both $S_1$ and $T_1$ of heptazine are almost dominated by the single-excitation configuration between

the highest occupied molecular orbital (HOMO) and the lowest unoccupied molecular orbital (LUMO)[15].

Figure 1c shows the statistics of the screened molecules as a function of $\Delta E_{ST}$ and $f$ calculated by TDDFT. A well-known trade-off between small $\Delta E_{ST}$ and large $f$ is evident from this particular dataset of heptazine analogues. Although balancing such a trade-off is a key concern in recent synthetic efforts on TADF materials, Fig. 1c demonstrates the optimal combinations of $\Delta E_{ST}$ and $f$ for which one parameter can no longer be improved without sacrificing the other. Figure 1d further visualizes the trade-off between $\Delta E_{ST}$ and $f$ for each fluorescence colour. The screening data suggest 5,264 promising candidates to show fluorescence across the entire visible spectrum, with $\Delta E_{ST} < 0.35$ eV and $f > 0.01$. Setting the range of the vertical $S_1$–$S_0$ energy gap to 2.70–2.85 eV for blue fluorescence further narrows down the candidates to 1,028 molecules, corresponding to 2.97% of all the screened molecules. We then assessed their synthetic feasibility and selected two heptazine analogues HzTFEX$_2$ and HzPipX$_2$ (Fig. 2a) for further evaluation. We note that these molecules recover $f$ while retaining small $\Delta E_{ST}$ ($f = 0.010$ and 0.015 and $\Delta E_{ST} = 210$ and 334 meV for HzTFEX$_2$ and HzPipX$_2$, respectively). This trend is consistent with the recent computational screening on heptazine analogues with asymmetrical substitutions[19].

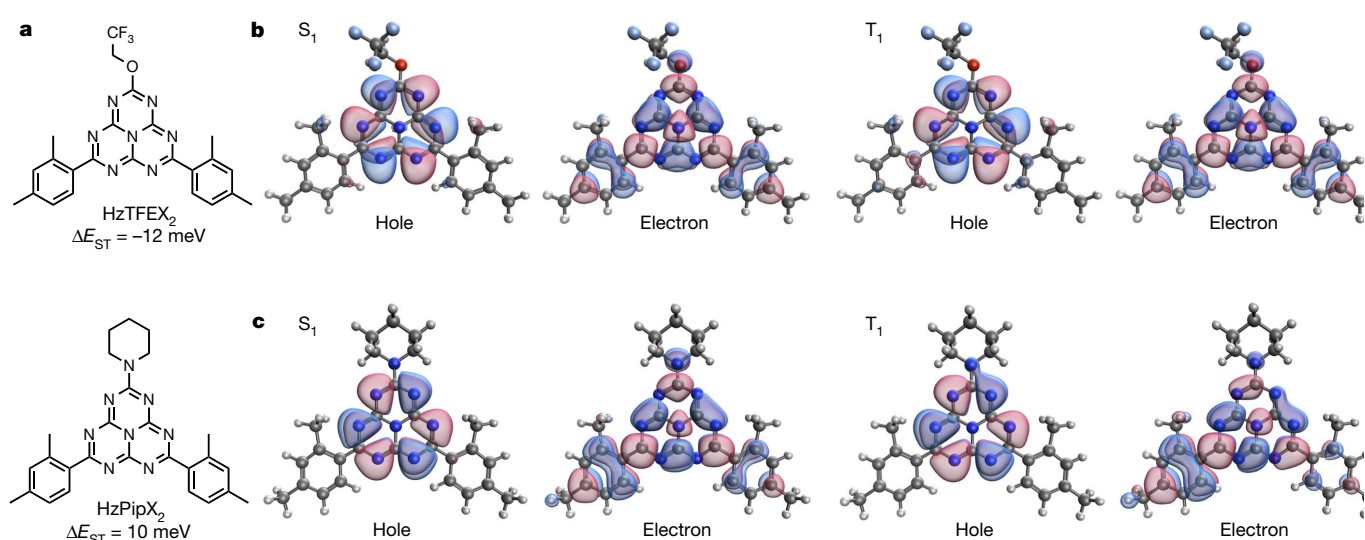

**Fig. 2 | Lead candidate molecules HzTFEX$_2$ and HzPipX$_2$. a**, Molecular structures of HzTFEX$_2$ and HzPipX$_2$ with $\Delta E_{ST}$ calculated by EOM-CCSD. **b,c**, Dominant pair of NTOs of $S_1$ and $T_1$ of HzTFEX$_2$ (**b**) and HzPipX$_2$ (**c**) for the EOM-CCSD calculations.

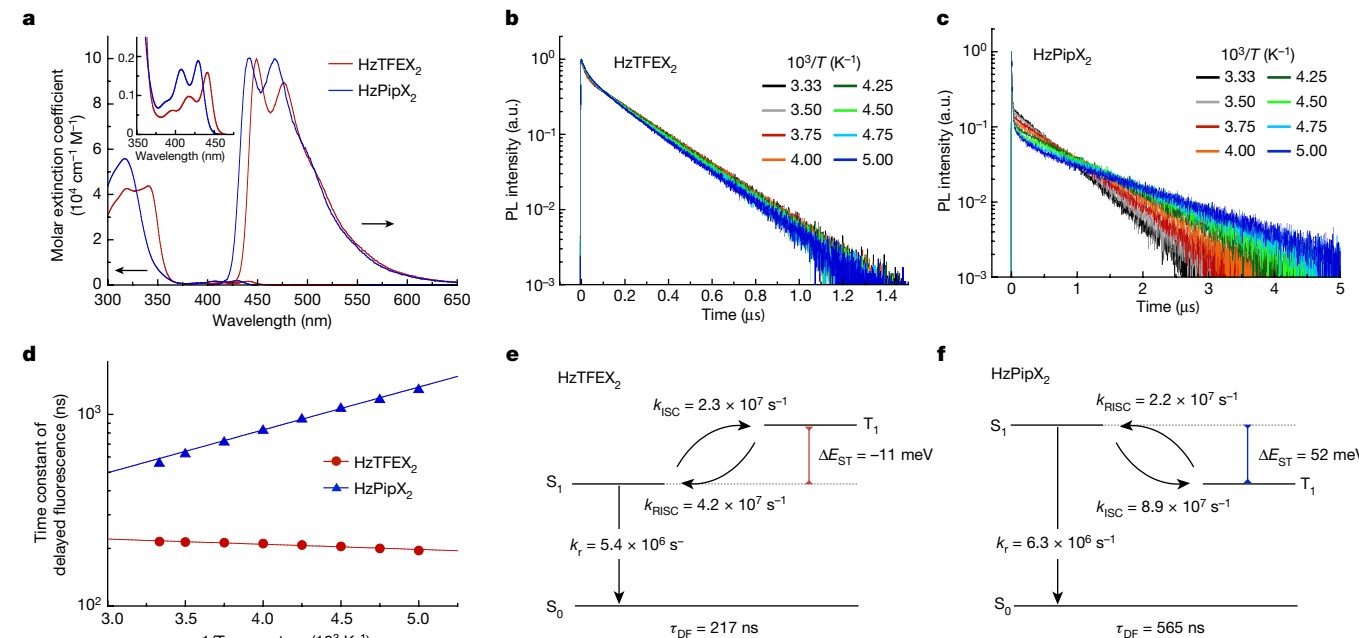

**Fig. 3 | Photophysical properties of HzTFEX$_2$ and HzPipX$_2$ in deaerated toluene solutions. a**, Steady-state absorption and PL spectra of HzTFEX$_2$ and HzPipX$_2$. The inset is the magnified view of the absorption spectra. **b,c**, Transient PL decays of HzTFEX$_2$ (**b**) and HzPipX$_2$ (**c**) at varying temperatures. **d**, Temperature dependence of $\tau_{DF}$ of HzTFEX$_2$ and HzPipX$_2$; the solid lines in **d** represent the fits of $\tau_{DF}$ to a single exponential in inverse temperature. **e,f**, Schematic diagram of the excited states and the associated transitions of HzTFEX$_2$ (**e**) and HzPipX$_2$ (**f**).

To examine whether HzTFEX$_2$ and HzPipX$_2$ could have negative $\Delta E_{ST}$, we computed their S$_1$ and T$_1$ by correlated wave function theories. Equation-of-motion coupled cluster with single and double excitation (EOM-CCSD)[29] calculations predict HzTFEX$_2$ to possess negative $\Delta E_{ST}$ of −12 meV, affirming its potential for exhibiting DFIST. In comparison, $\Delta E_{ST}$ calculated for HzPipX$_2$ remains at a positive value of 10 meV, which is comparable with those of the current state-of-the-art TADF materials[30–38]. Figure 2b,c shows the dominant pair of natural transition orbitals (NTOs)[39] for S$_1$ and T$_1$. In both molecules, the hole orbitals are exclusively localized on the peripherical six nitrogen atoms of the heptazine core, whereas the electron orbitals are localized on the central nitrogen atom and the carbon atoms of the core, as well as on the substituents. The spatial separation of these orbitals indicates that the exchange interaction is weak, resulting in nearly degenerate S$_1$ and T$_1$ in the single-excitation picture. Similar spatial separations of NTOs have also been found in the multi-resonant TADF materials, such as DABNA-1 (refs. [22,23]). In this situation, the stabilization of S$_1$ by including the double-excitation configurations becomes more dominant to determine the sign of $\Delta E_{ST}$. Indeed, S$_1$ of both molecules comprise double-excitation configurations with weights of around 1% described as the sum of the squares of the doubles amplitudes in EOM-CCSD, which are slightly higher than those of T$_1$. Two other wave-function-based calculations using second-order algebraic diagrammatic construction (ADC(2))[40] and complete active space with second-order perturbation theory (CASPT2)[41] further validate the inversion of S$_1$ and T$_1$ in HzTFEX$_2$ with calculated $\Delta E_{ST}$ of −34 meV and −184 meV, respectively. However, for HzPipX$_2$, the two methods also invert $\Delta E_{ST}$ (−12 meV with ADC(2) and −171 meV with CASPT2) as compared with the positive value of 10 meV predicted with EOM-CCSD (Supplementary Table 1). This variation in estimates of $\Delta E_{ST}$ highlights the current limitations of excited-state calculations and demands conclusive experimental evaluation. We note that $\Delta E_{ST}$ calculated by other second-order methods are given in the Supplementary Information, as well as the dependence of the choice of the guess orbitals and the size of the active space on the CASPT2 results.

HzTFEX$_2$ and HzPipX$_2$ were synthesized by nucleophilic aromatic substitution of 2,5,8-trichloroheptazine with corresponding alcohol or amine, followed by Friedel–Crafts reactions with *m*-xylene. The details of the synthesis and characterization are given in the Supplementary Information. The photophysical properties of the two molecules were evaluated in deaerated toluene solutions (Fig. 3a and Extended Data Table 1). The steady-state absorption spectra of HzTFEX$_2$ and HzPipX$_2$ comprise the lowest-energy absorption band centred at 441 nm and 429 nm, respectively, with small molar absorption coefficients on the order of $10^3$ M$^{-1}$ cm$^{-1}$, reflecting the spatial separation between the hole and electron NTOs computed for S$_1$ of each molecule. On photoexcitation, HzTFEX$_2$ exhibits blue emission with a peak wavelength ($\lambda_{PL}$) of 449 nm and a photoluminescence (PL) quantum yield ($\Phi_{PL}$) of 74%, whereas slightly blue-shifted $\lambda_{PL}$ of 442 nm and similar $\Phi_{PL}$ of 67% are observed for HzPipX$_2$. These energy differences in absorption and emission are also predicted by TDDFT calculations and are attributed to the stronger electron-donating effect of the piperidyl group in HzPipX$_2$ than that of 2,2,2-trifluoroethoxy group in HzTFEX$_2$. In aerated toluene solutions, $\Phi_{PL}$ of HzTFEX$_2$ and HzPipX$_2$ decrease to 54% and 37%, respectively. Because atmospheric O$_2$ can quench molecular triplet excited states and the change in $\Phi_{PL}$ is reversible, we ascribe the blue emissions of the two molecules, at least partially, to delayed fluorescence through forward intersystem crossing (ISC) and RISC between S$_1$ and T$_1$. This assumption is supported by transient absorption decay measurements on HzTFEX$_2$, which scrutinized ISC from S$_1$ to T$_1$ as the signal decay of S$_1$ at 700 nm and the signal growth of T$_1$ at 1,600 nm, followed by the persistent signal decays of both S$_1$ and T$_1$ (Extended Data Fig. 1). We also note that both decays have similar time constants (223 ns for S$_1$ and 210 ns for T$_1$), indicating the steady-state condition with the constant population ratio maintained by ISC and RISC.

To show the excited-state kinetics of the two molecules in detail, we performed transient PL decay measurements at varying temperatures (Fig. 3b,c and Supplementary Fig. 3 for the log–log representation). Both molecules exhibit biexponential transient PL decays, which comprise nanosecond-order prompt fluorescence followed by sub-microsecond delayed fluorescence with temperature-dependent time constants.

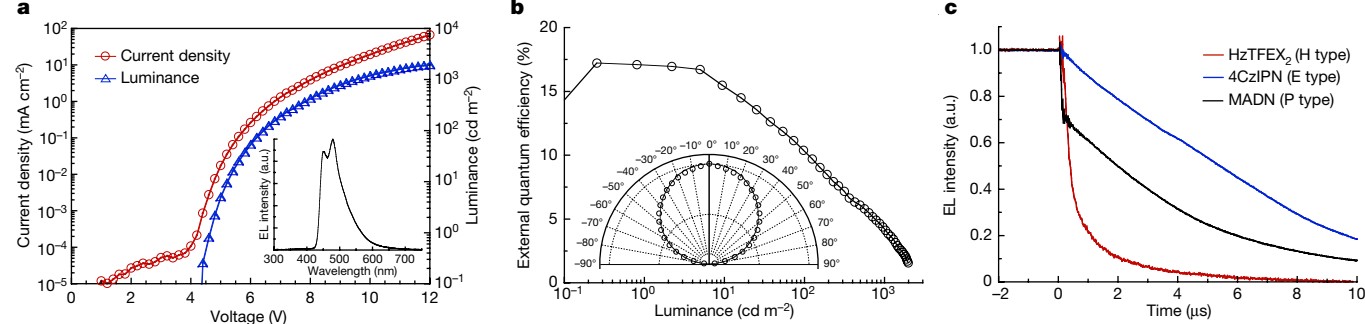

**Fig. 4 | OLED performance. a,b,** Current density–voltage–luminance characteristics (**a**) and external quantum efficiency–luminance characteristics (**b**) of the fabricated OLED using HzTFEX$_2$; the inset in **a** shows the EL spectra measured at 1.0 mA and the inset in **b** represents the viewing-angle dependence of the luminance, which is almost consistent with the Lambertian distribution. **c,** Transient EL decays of the OLEDs using HzTFEX$_2$, 4CzIPN and MADN, respectively, measured in pulse operation with square-wave voltages of 8 V and −4 V.

Remarkably, the time constant of delayed fluorescence ($\tau_{DF}$) of HzTFEX$_2$ gradually decreases from 217 ns to 195 ns with decreasing temperature from 300 K to 200 K (Fig. 3d). This anomalous temperature dependence of $\tau_{DF}$ indicates that S$_1$ lies energetically below T$_1$, for which lowering the temperature shifts the steady-state population towards emissive S$_1$ relative to dark T$_1$ and thus accelerates the delayed fluorescence (that is, decreases $\tau_{DF}$). In comparison, $\tau_{DF}$ of HzPipX$_2$ increases from 565 ns to 1,372 ns by the same temperature decrease, as has been similarly observed in conventional TADF materials[5–7]. It is worth noting that $\tau_{DF}$ of HzTFEX$_2$ is much shorter than emission time constants ever reported for TADF materials[30–38] and phosphorescent materials[2–4] used for efficient OLEDs, which are typically in the microsecond range.

We further analysed the temperature-dependent PL decay kinetics with the underlying rate equation. In the absence of phosphorescence and non-radiative decay of T$_1$ to S$_0$, the rate equation for the populations of S$_1$ and T$_1$ is given by

$$\frac{d}{dt}\begin{pmatrix} S_1 \\ T_1 \end{pmatrix} = \begin{pmatrix} -(k_r + k_{nr} + k_{ISC}) & k_{RISC} \\ k_{ISC} & -k_{RISC} \end{pmatrix}\begin{pmatrix} S_1 \\ T_1 \end{pmatrix} \quad (1)$$

in which $k_r$, $k_{nr}$, $k_{ISC}$ and $k_{RISC}$ are the rate constants of radiative decay of S$_1$ to S$_0$, non-radiative decay of S$_1$ to S$_0$, ISC of S$_1$ to T$_1$ and RISC of T$_1$ to S$_1$, respectively. By numerically fitting equation (1) to the PL decay data at 300 K, we found that RISC is faster than ISC in HzTFEX$_2$ ($k_{RISC} = 4.2 \times 10^7$ s$^{-1}$ versus $k_{ISC} = 2.3 \times 10^7$ s$^{-1}$), whereas RISC is slower than ISC in HzPipX$_2$ ($k_{RISC} = 2.2 \times 10^7$ s$^{-1}$ versus $k_{ISC} = 8.9 \times 10^7$ s$^{-1}$) (Fig. 3e,f). These parameters simulate that the population of T$_1$ is lower than that of S$_1$ in HzTFEX$_2$ under the steady-state condition, indicating that S$_1$ lies energetically below T$_1$ (Extended Data Fig. 2). Furthermore, the temperature dependence of $k_{ISC}$ and $k_{RISC}$ follows the Arrhenius equation, $k = A\exp(-E_a/k_B T)$, in which $k$ is the rate constant, $A$ is the pre-exponential factor, $E_a$ is the activation energy, $k_B$ is the Boltzmann constant and $T$ is the absolute temperature (Extended Data Fig. 3). The best-fit parameters of the Arrhenius equation yield the activation energies of ISC and RISC ($E_{a,ISC}$ and $E_{a,RISC}$) (Extended Data Table 1). Subtracting $E_{a,ISC}$ from $E_{a,RISC}$, we determined $\Delta E_{ST}$ of HzTFEX$_2$ to be −11 ± 2 meV, which is in marked contrast to positive $\Delta E_{ST}$ ever observed in numerous molecules, as well as in HzPipX$_2$ ($\Delta E_{ST}$ = 52 ±1 meV). We note that the change in $k_r + k_{nr}$ at varying temperatures is negligible compared with those in $k_{ISC}$ and $k_{RISC}$ (Supplementary Fig. 4) and thus the decreasing trend of $\tau_{DF}$ of HzTFEX$_2$ is more reasonably attributed to the inverted S$_1$ and T$_1$. The negative $\Delta E_{ST}$ of HzTFEX$_2$ is retained in a solid-state host matrix (see Extended Data Fig. 4 and the Supplementary Information for details).

Having experimentally determined negative $\Delta E_{ST}$, we conclude that HzTFEX$_2$ exhibits DFIST. Further synthetic efforts replacing the xylyl groups in HzTFEX$_2$ with either phenyl or tolyl groups led to HzTFEP$_2$ and HzTFET$_2$, which similarly show DFIST with measured $\Delta E_{ST}$ of −14 ± 3 meV and −13 ± 3 meV, respectively (see Extended Data Table 1 and Supplementary Fig. 5 for details), indicating the potential of heptazines for further developing efficient DFIST materials. In common with the three materials, ISC from S$_1$ to T$_1$ competes with the inherently slow radiative decay of heptazines, followed by faster RISC, leading to a significant S$_1$ population relative to T$_1$ and sub-microsecond DFIST. Thus, we propose to refer to the present type of emissions as 'H (heptazine)-type delayed fluorescence' by analogy with 'E (eosin)-type delayed fluorescence' referred to as TADF[42] and 'P (pyren)-type delayed fluorescence' involving triplet–triplet annihilation[43].

Finally, we evaluated the electroluminescence (EL) properties of HzTFEX$_2$ in OLEDs fabricated by thermal evaporation. The details of the fabrication procedures and the device structures are given in the Supplementary Information. Figure 4a,b shows the EL spectra, current density–voltage–luminance characteristics and external quantum efficiency–luminance characteristics of the OLED. Intense blue EL originating from HzTFEX$_2$ was observed with spectral peak wavelengths ($\lambda_{EL}$) at 450 nm and 479 nm and Commission internationale de l'éclairage (CIE) coordinates of (0.17, 0.24). The maximum external quantum efficiency reached 17%, corresponding to the internal quantum efficiency of 80% for a bottom-emission OLED with a typical light-outcoupling efficiency of 20%[44]. We note that the viewing-angle dependence of the luminance followed the Lambertian distribution (Fig. 4b inset), ensuring accurate estimation of the external quantum efficiency from the forward emission. Remarkably, HzTFEX$_2$ exhibited fast transient EL decay, reflecting the sub-microsecond H-type delayed fluorescence (Fig. 4c). In comparison, much slower transient EL decays were observed for E-type delayed fluorescence of 2,4,5,6-tetra(carbazol-9-yl)isophthalonitrile (4CzIPN)[6] and P-type delayed fluorescence of 2-methyl-9,10-bis(naphthalen-2-yl) anthracene (MADN)[45], although the EL of MADN initially decayed faster by the prompt fluorescence solely from S$_1$ (ref. [46]). It is thus evident that the fast triplet harvesting of HzTFEX$_2$ with negative $\Delta E_{ST}$ can be retained even in actual OLEDs. Although the efficiency roll-off is still marked in this preliminary device concerning the large hole injection barrier caused by the high ionization potential of HzTFEX$_2$ (6.3 eV), we anticipate that further optimization of molecular design will address this issue and allow a conclusive exploration of the effects of negative $\Delta E_{ST}$ on efficiency roll-off and device stability.

In conclusion, we have demonstrated fluorescent heptazine molecules that possess negative $\Delta E_{ST}$. We observed their blue delayed fluorescence in both PL and EL with anomalous features: (1) the very short decay time constants ($\tau_{DF} \approx 0.2$ μs), (2) the decreasing trend of $\tau_{DF}$ with decreasing temperature and (3) the rate inversion of RISC and ISC ($k_{RISC} > k_{ISC}$). These features indeed arise from negative $\Delta E_{ST}$ and led to the terminology 'delayed fluorescence from inverted singlet and triplet

excited states (DFIST)' or 'H (heptazine)-type delayed fluorescence'. We predict that further development of DFIST materials will offer stable and efficient OLEDs based on the fast triplet-to-singlet downconversion, with great implications for displays, lighting and lasers.

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

## Methods

### Quantum-chemical calculations

For the 34,596 heptazine molecules, the $T_1$ geometries were optimized using spin-unrestricted DFT with the LC-BLYP functional and the 6-31G basis set. Vibrational frequency analysis for HzTFEX$_2$, HzPipX$_2$, HzTFEP$_2$ and HzTFET$_2$ gave no imaginary frequencies at the same level of theory. The vertical excitation energies of $S_1$ and $T_1$ were calculated using liner-response TDDFT with the LC-BLYP functional and the 6-31G(d) basis set within the Tamm–Dancoff approximation. The range-separation parameter of the LC-BLYP functional was non-empirically optimized to 0.18 bohr$^{-1}$ to minimize the difference between the energy of the HOMO and the ionization potential of the neutral system and the difference between the energy of the HOMO of the radical anion system and the electron affinity of the neutral system[47] of 2,5,8-triphenylheptazine. The $T_1$ geometries of HzTFEX$_2$ and HzPipX$_2$ were also optimized using spin-unrestricted second-order Møller–Plesset perturbation theory (MP2) with the correlation consistent cc-pVDZ basis set. At the MP2 geometries of HzTFEX$_2$ and HzPipX$_2$, the vertical excitation energies of $S_1$ and $T_1$ were calculated using EOM-CCSD[29], ADC(2)[40] and CASPT2[41] with the cc-pVDZ basis set. The CASPT2 calculations were performed with the fully internally contracted scheme over the state-averaged complete active space self-consistent field (CASSCF) wavefunctions with the active space of 12 electrons and 12 orbitals using the resolution of identity approximation with the auxiliary fitting basis set. The DFT, TDDFT, MP2 and EOM-CCSD calculations were performed using the Gaussian 16 Rev C.01 program. The ADC(2) calculations were performed using the Q-Chem 5.3.0 program. The CASSCF and CASPT2 calculations were performed using the Orca 4.2.1 program.

### Materials and synthesis

Commercially available reagents and solvents were used without further purification unless otherwise noted. 4CzIPN and MADN were purchased from Luminescence Technology Corporation and e-Ray Optoelectronics Technology, respectively. The synthetic procedures and characterization data of the heptazine molecules are detailed in the Supplementary Information.

### Photophysical measurements

Steady-state ultraviolet–visible absorption spectra were recorded on a Shimadzu UV-3600i Plus spectrophotometer. Steady-state PL spectra were acquired on a HORIBA FL3 spectrofluorometer with 370-nm photoexcitation from a Xe arc lamp. The absolute PL quantum yields were determined using a Hamamatsu Photonics C9920 integrated sphere system with 370-nm excitation from a Xe arc lamp. Transient absorption decay measurements were performed by a randomly interleaved plus train method[48] on a UNISOKU picoTAS system with a 355-nm Q-switched laser pump source (pulse width <350 ps) and a supercontinuum white probe source (pulse width <100 ps). Transient PL decay measurements were performed by time-correlated single-photon counting on a HORIBA FL3 spectrofluorometer with a 370-nm LED pump source (pulse width <1.2 ns) and a UNISOKU CoolSpek cryostat using liquid nitrogen as the coolant. Ionization potentials were determined using a RIKEN KEIKI AC-3 ultraviolet photoelectron yield spectrometer.

### Analysis of transient PL decay kinetics

The time constants of prompt and delayed fluorescence ($\tau_{PF}$ and $\tau_{DF}$) were determined by biexponential decay fitting and deconvolution with the instrument response function. It is common when determining the rate constants of the transitions involved in TADF to assume $k_{ISC} \gg k_{RISC}$ such that the contribution of RISC to the prompt fluorescence is negligible[49]. However, this assumption does not hold true for DFIST materials with $k_{RISC} > k_{ISC}$. Thus, $k_r + k_{nr}$, $k_{ISC}$ and $k_{RISC}$ were determined without assuming $k_{ISC} \gg k_{RISC}$ by fitting the $S_1$ population in equation (1) to the transient PL decay data using the scipy.integrate.odeint and scipy.optimize.curve_fit functions in Python 3.7[50]. $k_r$ and $k_{nr}$ were determined from $\Phi_{PL} = k_r/(k_r + k_{nr})$ assuming negligible non-radiative decay of $T_1$ to $S_0$. Activation energies of ISC and RISC ($E_{a,ISC}$ and $E_{a,RISC}$) were determined by fitting the Arrhenius equation to the temperature dependence of $k_{ISC}$ and $k_{RISC}$, respectively. $\Delta E_{ST}$ was determined by subtracting $E_{a,ISC}$ from $E_{a,RISC}$.

### OLED fabrication and evaluation

The fabrication procedures of OLEDs are detailed in the Supplementary Information. EL spectra were recorded using a Hamamatsu Photonics PMA-12 photonic multichannel analyser. Current density–voltage–luminance characteristics were measured using a Konica Minolta CS-200 luminance meter and a Keithley 2400 source meter. The viewing-angle dependence of luminance was measured using a home-build spectro-goniometer with a Konica Minolta CS-2000 spectroradiometer. Transient EL decays measurements were performed using a home-build set-up with a Hamamatsu Photonics H7826 silicon photomultiplier tube (time response = 1.5 ns) and an Agilent 33220A function generator for pulse OLED operation (square-wave voltages = 8, −4 V and frequency = 2 kHz).

### Reporting summary

Further information on research design is available in the Nature Research Reporting Summary linked to this article.

### Data availability

The data underlying this article are available at https://doi.org/10.6084/m9.figshare.20058977.

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

**Acknowledgements** This work was supported in part by an Industrial Technology Research Grant for Young Researchers from the New Energy and Industrial Technology Development Organization (NEDO) (grant no. 09151455 to Y.-J.P.), JST PRESTO (grant nos. 13417316 to Y.-J.P. and JPMJPR17N1 to N.A.) and JSPS KAKENHI (grant nos. 24685029, 17H03103, 20H02554 to Y.-J.P. and 20K15252, 21H05413, 22H02051 to N.A.). The authors thank M. Kim at RIKEN CEMS, T. Chiba, and M. Hirasawa at Yamagata University for their support with the OLED fabrication and evaluation. The computations were partially performed using the HOKUSAI Big Waterfall system at RIKEN.

**Author contributions** N.A. and D.M. conceived the project. N.A., Y.H. and S.M. conceived the procedures of the computational calculations. N.A., Y.-J.P., Y.H., K.N. and S.M. performed the computational calculations. N.A., H.I. and Y.-J.P. performed the photophysical measurements. N.A., Y.K. and Y.-J.P. fabricated and evaluated the OLEDs. B.D., R.I., H.I. and A.N. performed synthetic experiments and assisted the characterization of the synthesized compounds and analysis of data. N.A., Y.-J.P., F.A. and D.M. designed the experiments and analysed the data. N.A., Y.-J.P. and D.M. wrote the manuscript.

**Competing interests** The authors declare no competing interests.

**Additional information**
**Correspondence and requests for materials** should be addressed to Naoya Aizawa, Yong-Jin Pu or Daigo Miyajima.

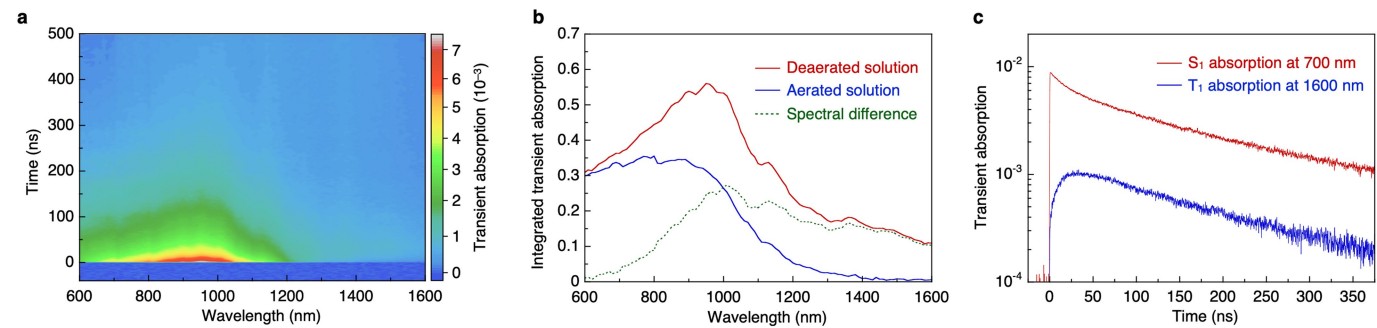

**Extended Data Fig. 1 | Transient absorption data of HzTFEX$_2$. a**, Transient absorption of HzTFEX$_2$ as a function of wavelength and time in a deaerated toluene solution. **b**, Integrated transient absorption spectra of HzTFEX$_2$ over 0–500 ns in deaerated and aerated toluene solutions; the dashed green dashed line represents their spectral difference and mainly corresponds to the transient absorption of T$_1$. **c**, Transient absorption decays of S$_1$ and T$_1$ monitored at 700 nm and 1,600 nm, respectively.

**a**

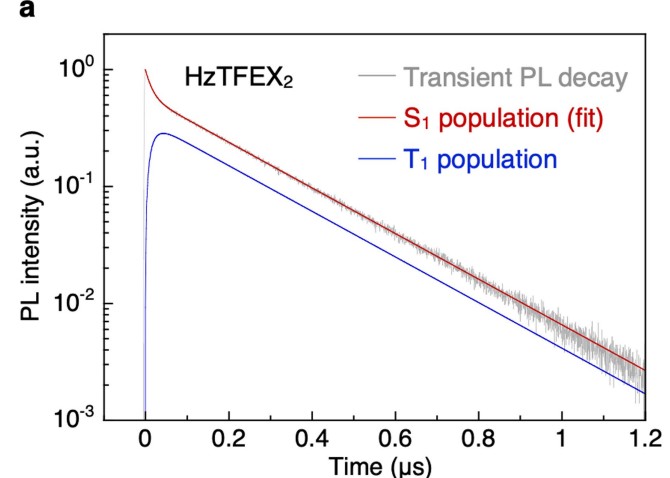

**b**

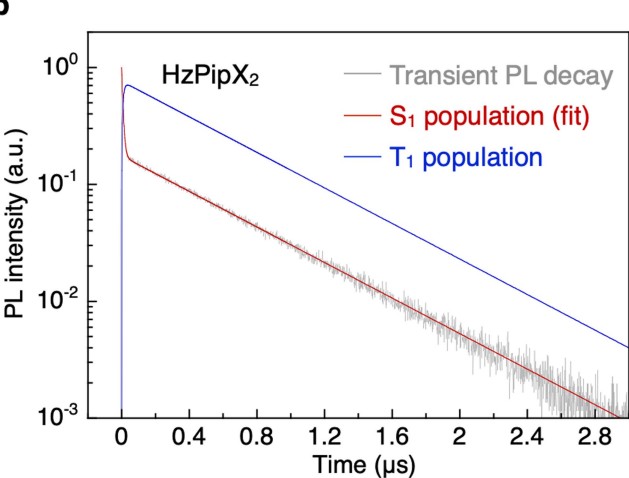

**Extended Data Fig. 2 | Analysis of the transient PL decays of HzTFEX$_2$ and HzPipX$_2$. a,b**, Fit (solid red line) of the S$_1$ population in equation (1) to the transient PL decay of HzTFEX$_2$ (**a**) and HzPipX$_2$ (**b**) in deaerated toluene solutions at 300 K. The solid blue line represents the T$_1$ population simulated by the best-fit parameters $k_r + k_{nr}$, $k_{ISC}$ and $k_{RISC}$ in equation (1).

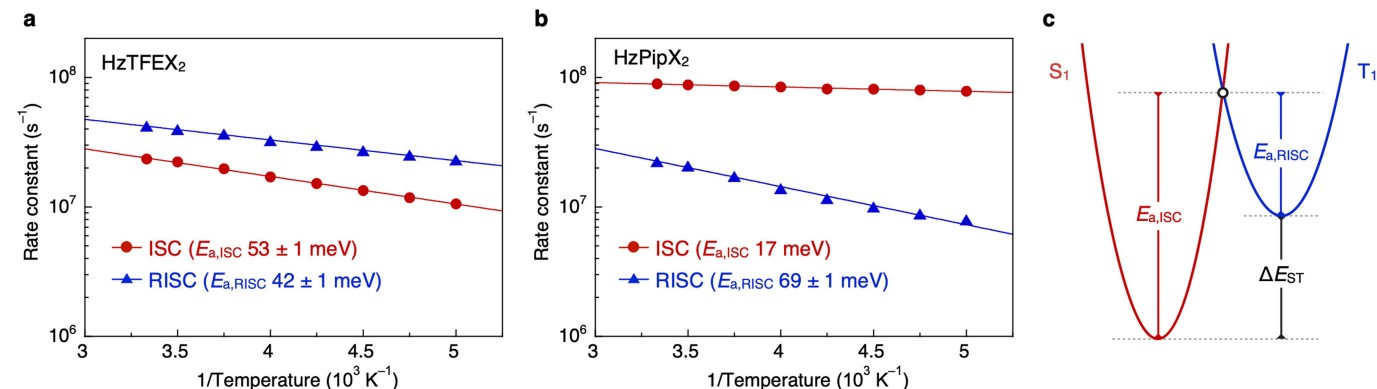

**Extended Data Fig. 3 | $k_{ISC}$ and $k_{RISC}$ of HzTFEX₂ and HzPipX₂. a,b**, Temperature dependence of $k_{ISC}$ and $k_{RISC}$ of HzTFEX₂ (**a**) and HzPipX₂ (**b**) in deaerated toluene. The solid lines in **a** and **b** represent the fits of the plots to the Arrhenius equation. The error bars of the plots in **a** and **b** are smaller than the plot size. **c**, Schematic diagram of the potential energy surfaces of S₁ and T₁ and the activation energies of ISC and RISC.

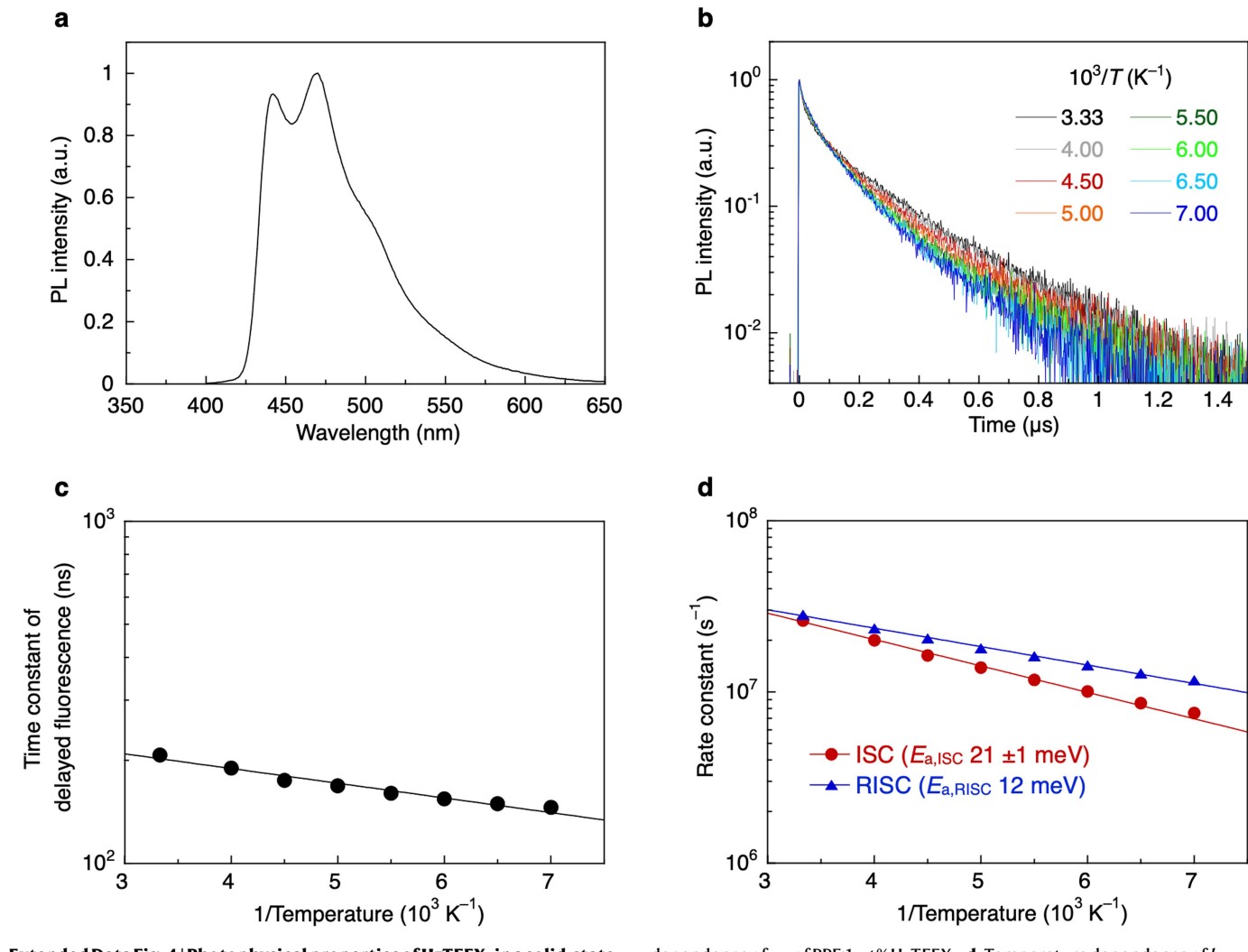

**Extended Data Fig. 4 | Photophysical properties of HzTFEX$_2$ in a solid-state host matrix. a**, Steady-state PL spectra of a thin film of bis(diphenylphosphoryl) dibenzo[$b,d$]furan (PPF):1 wt% HzTFEX$_2$. **b**, Transient PL decays of PPF:1 wt% HzTFEX$_2$ at varying temperatures under a N$_2$ atmosphere. **c**, Temperature dependence of $\tau_{DF}$ of PPF:1 wt% HzTFEX$_2$. **d**, Temperature dependence of $k_{ISC}$ and $k_{RISC}$ of PPF:1 wt% HzTFEX$_2$. The solid lines in **d** represent the fits of the plots to the Arrhenius equation.

**Extended Data Table 1 | Photophysical properties of HzTFEX$_2$ and HzPipX$_2$ in deaerated toluene solutions**

| Emitter | $\lambda_{PL}$ (nm)[a] | $\Phi_{PL}$ (%)[b] | $\tau_{PF}$ (ns)[c] | $\tau_{DF}$ (ns)[d] | $k_r$ (s$^{-1}$)[e] | $k_{nr}$ (s$^{-1}$)[f] | $k_{ISC}$ (s$^{-1}$)[g] | $k_{RISC}$ (s$^{-1}$)[h] | $E_{a,ISC}$ (meV)[i] | $E_{a,RISC}$ (meV)[j] | $\Delta E_{ST}$ (meV)[k] |
|---|---|---|---|---|---|---|---|---|---|---|---|
| HzTFEX$_2$ | 449, 476 | 74 | 14 | 217 | $5.4 \times 10^6$ | $1.9 \times 10^6$ | $2.3 \times 10^7$ | $4.2 \times 10^7$ | $53 \pm 1$ | $42 \pm 1$ | $-11 \pm 2$ |
| HzPipX$_2$ | 442, 467 | 67 | 7.9 | 565 | $6.3 \times 10^6$ | $3.1 \times 10^6$ | $8.9 \times 10^7$ | $2.2 \times 10^7$ | 17 | $69 \pm 1$ | $52 \pm 1$ |
| HzTFEP$_2$ | 454, 483 | 44 | 35 | 288 | $3.1 \times 10^6$ | $4.0 \times 10^6$ | $1.4 \times 10^7$ | $1.8 \times 10^7$ | $31 \pm 2$ | $17 \pm 1$ | $-14 \pm 3$ |
| HzTFET$_2$ | 451, 479 | 42 | 23 | 246 | $2.9 \times 10^6$ | $4.0 \times 10^6$ | $1.6 \times 10^7$ | $2.7 \times 10^7$ | $44 \pm 2$ | $31 \pm 1$ | $-13 \pm 3$ |

[a]Photoluminescence (PL) peak wavelength. [b]PL quantum yield. [c]Time constant of prompt fluorescence. [d]Time constant of delayed fluorescence. [e]Rate constant of radiative decay of the lowest-energy excited state (S$_1$) to the ground state (S$_0$). [f]Rate constant of non-radiative decay of S$_1$ to S$_0$. [g]Rate constant of intersystem crossing (ISC) of S$_1$ to the lowest-energy triplet excited state (T$_1$). [h]Rate constant of reverse intersystem crossing (RISC) of T$_1$ to S$_1$. [i]Activation energy of ISC. [j]Activation energy of RISC. [k]Energy gap between S$_1$ and T$_1$.

# Reporting Summary

## Statistics

For all statistical analyses, confirm that the following items are present in the figure legend, table legend, main text, or Methods section.

| n/a | Confirmed | |
|---|---|---|
| ☐ | ☒ | The exact sample size ($n$) for each experimental group/condition, given as a discrete number and unit of measurement |
| ☐ | ☒ | A statement on whether measurements were taken from distinct samples or whether the same sample was measured repeatedly |
| ☐ | ☒ | The statistical test(s) used AND whether they are one- or two-sided<br>*Only common tests should be described solely by name; describe more complex techniques in the Methods section.* |
| ☐ | ☒ | A description of all covariates tested |
| ☐ | ☒ | A description of any assumptions or corrections, such as tests of normality and adjustment for multiple comparisons |
| ☐ | ☒ | A full description of the statistical parameters including central tendency (e.g. means) or other basic estimates (e.g. regression coefficient) AND variation (e.g. standard deviation) or associated estimates of uncertainty (e.g. confidence intervals) |
| ☐ | ☒ | For null hypothesis testing, the test statistic (e.g. $F$, $t$, $r$) with confidence intervals, effect sizes, degrees of freedom and $P$ value noted<br>*Give P values as exact values whenever suitable.* |
| ☐ | ☒ | For Bayesian analysis, information on the choice of priors and Markov chain Monte Carlo settings |
| ☐ | ☒ | For hierarchical and complex designs, identification of the appropriate level for tests and full reporting of outcomes |
| ☐ | ☒ | Estimates of effect sizes (e.g. Cohen's $d$, Pearson's $r$), indicating how they were calculated |

*Our web collection on statistics for biologists contains articles on many of the points above.*

## Software and code

Policy information about availability of computer code

| | |
|---|---|
| Data collection | Quantum chemical calculations were performed using Gaussian 16 RevC.01, Q-Chem 5.3.0, Orca 4.2.1, and MOLPRO 2019.2. |
| Data analysis | The numerical fit of the S1 population in Eq. (1) to the transient PL decay data was performed using the scipy.integrate.odeint and scipy.optimize.curve_fit functions in Python 3.7. |

For manuscripts utilizing custom algorithms or software that are central to the research but not yet described in published literature, software must be made available to editors and reviewers. We strongly encourage code deposition in a community repository (e.g. GitHub). See the Nature Portfolio guidelines for submitting code & software for further information.

## Data

Policy information about availability of data

All manuscripts must include a data availability statement. This statement should provide the following information, where applicable:
- Accession codes, unique identifiers, or web links for publicly available datasets
- A description of any restrictions on data availability
- For clinical datasets or third party data, please ensure that the statement adheres to our policy

The data underlying this article are available at https://doi.org/10.6084/m9.figshare.20058977. (The DOI will become active when the paper is published.)

# Human research participants

Policy information about studies involving human research participants and Sex and Gender in Research.

| | |
|---|---|
| Reporting on sex and gender | *Use the terms sex (biological attribute) and gender (shaped by social and cultural circumstances) carefully in order to avoid confusing both terms. Indicate if findings apply to only one sex or gender; describe whether sex and gender were considered in study design whether sex and/or gender was determined based on self-reporting or assigned and methods used. Provide in the source data disaggregated sex and gender data where this information has been collected, and consent has been obtained for sharing of individual-level data; provide overall numbers in this Reporting Summary. Please state if this information has not been collected. Report sex- and gender-based analyses where performed, justify reasons for lack of sex- and gender-based analysis.* |
| Population characteristics | *Describe the covariate-relevant population characteristics of the human research participants (e.g. age, genotypic information, past and current diagnosis and treatment categories). If you filled out the behavioural & social sciences study design questions and have nothing to add here, write "See above."* |
| Recruitment | *Describe how participants were recruited. Outline any potential self-selection bias or other biases that may be present and how these are likely to impact results.* |
| Ethics oversight | *Identify the organization(s) that approved the study protocol.* |

Note that full information on the approval of the study protocol must also be provided in the manuscript.

# Field-specific reporting

Please select the one below that is the best fit for your research. If you are not sure, read the appropriate sections before making your selection.

☐ Life sciences ☐ Behavioural & social sciences ☐ Ecological, evolutionary & environmental sciences

For a reference copy of the document with all sections, see nature.com/documents/nr-reporting-summary-flat.pdf

# Life sciences study design

All studies must disclose on these points even when the disclosure is negative.

| | |
|---|---|
| Sample size | *Describe how sample size was determined, detailing any statistical methods used to predetermine sample size OR if no sample-size calculation was performed, describe how sample sizes were chosen and provide a rationale for why these sample sizes are sufficient.* |
| Data exclusions | *Describe any data exclusions. If no data were excluded from the analyses, state so OR if data were excluded, describe the exclusions and the rationale behind them, indicating whether exclusion criteria were pre-established.* |
| Replication | *Describe the measures taken to verify the reproducibility of the experimental findings. If all attempts at replication were successful, confirm this OR if there are any findings that were not replicated or cannot be reproduced, note this and describe why.* |
| Randomization | *Describe how samples/organisms/participants were allocated into experimental groups. If allocation was not random, describe how covariates were controlled OR if this is not relevant to your study, explain why.* |
| Blinding | *Describe whether the investigators were blinded to group allocation during data collection and/or analysis. If blinding was not possible, describe why OR explain why blinding was not relevant to your study.* |

# Behavioural & social sciences study design

All studies must disclose on these points even when the disclosure is negative.

| | |
|---|---|
| Study description | *Briefly describe the study type including whether data are quantitative, qualitative, or mixed-methods (e.g. qualitative cross-sectional, quantitative experimental, mixed-methods case study).* |
| Research sample | *State the research sample (e.g. Harvard university undergraduates, villagers in rural India) and provide relevant demographic information (e.g. age, sex) and indicate whether the sample is representative. Provide a rationale for the study sample chosen. For studies involving existing datasets, please describe the dataset and source.* |
| Sampling strategy | *Describe the sampling procedure (e.g. random, snowball, stratified, convenience). Describe the statistical methods that were used to predetermine sample size OR if no sample-size calculation was performed, describe how sample sizes were chosen and provide a rationale for why these sample sizes are sufficient. For qualitative data, please indicate whether data saturation was considered, and what criteria were used to decide that no further sampling was needed.* |

| | |
|---|---|
| Data collection | *Provide details about the data collection procedure, including the instruments or devices used to record the data (e.g. pen and paper, computer, eye tracker, video or audio equipment) whether anyone was present besides the participant(s) and the researcher, and whether the researcher was blind to experimental condition and/or the study hypothesis during data collection.* |
| Timing | *Indicate the start and stop dates of data collection. If there is a gap between collection periods, state the dates for each sample cohort.* |
| Data exclusions | *If no data were excluded from the analyses, state so OR if data were excluded, provide the exact number of exclusions and the rationale behind them, indicating whether exclusion criteria were pre-established.* |
| Non-participation | *State how many participants dropped out/declined participation and the reason(s) given OR provide response rate OR state that no participants dropped out/declined participation.* |
| Randomization | *If participants were not allocated into experimental groups, state so OR describe how participants were allocated to groups, and if allocation was not random, describe how covariates were controlled.* |

# Ecological, evolutionary & environmental sciences study design

All studies must disclose on these points even when the disclosure is negative.

| | |
|---|---|
| Study description | *Briefly describe the study. For quantitative data include treatment factors and interactions, design structure (e.g. factorial, nested, hierarchical), nature and number of experimental units and replicates.* |
| Research sample | *Describe the research sample (e.g. a group of tagged Passer domesticus, all Stenocereus thurberi within Organ Pipe Cactus National Monument), and provide a rationale for the sample choice. When relevant, describe the organism taxa, source, sex, age range and any manipulations. State what population the sample is meant to represent when applicable. For studies involving existing datasets, describe the data and its source.* |
| Sampling strategy | *Note the sampling procedure. Describe the statistical methods that were used to predetermine sample size OR if no sample-size calculation was performed, describe how sample sizes were chosen and provide a rationale for why these sample sizes are sufficient.* |
| Data collection | *Describe the data collection procedure, including who recorded the data and how.* |
| Timing and spatial scale | *Indicate the start and stop dates of data collection, noting the frequency and periodicity of sampling and providing a rationale for these choices. If there is a gap between collection periods, state the dates for each sample cohort. Specify the spatial scale from which the data are taken* |
| Data exclusions | *If no data were excluded from the analyses, state so OR if data were excluded, describe the exclusions and the rationale behind them, indicating whether exclusion criteria were pre-established.* |
| Reproducibility | *Describe the measures taken to verify the reproducibility of experimental findings. For each experiment, note whether any attempts to repeat the experiment failed OR state that all attempts to repeat the experiment were successful.* |
| Randomization | *Describe how samples/organisms/participants were allocated into groups. If allocation was not random, describe how covariates were controlled. If this is not relevant to your study, explain why.* |
| Blinding | *Describe the extent of blinding used during data acquisition and analysis. If blinding was not possible, describe why OR explain why blinding was not relevant to your study.* |

Did the study involve field work? ☐ Yes ☒ No

# Reporting for specific materials, systems and methods

We require information from authors about some types of materials, experimental systems and methods used in many studies. Here, indicate whether each material, system or method listed is relevant to your study. If you are not sure if a list item applies to your research, read the appropriate section before selecting a response.

## Materials & experimental systems

| n/a | Involved in the study |
|-----|----------------------|
| ☒ | ☐ Antibodies |
| ☒ | ☐ Eukaryotic cell lines |
| ☒ | ☐ Palaeontology and archaeology |
| ☒ | ☐ Animals and other organisms |
| ☒ | ☐ Clinical data |
| ☒ | ☐ Dual use research of concern |

## Methods

| n/a | Involved in the study |
|-----|----------------------|
| ☒ | ☐ ChIP-seq |
| ☒ | ☐ Flow cytometry |
| ☒ | ☐ MRI-based neuroimaging |

