## [Peer Review File · Nature]

Manuscript Title: Delayed Fluorescence from Inverted Singlet and Triplet Excited States

Reviewer Comments & Author Rebuttals

Reviewer Reports on the Initial Version:

Referees' comments:

Referee #1 (Remarks to the Author):

The interconversion of triplet to singlet excitons (formed upon a 3:1 ratio following the spin statistics) by a Reverse Intersystem Crossing Process (RISC) has received much attention as a promising mechanism to maximize the efficiency of light-emission, thus minimizing the loss of common fluorescence (energy decay from the lowest spin-singlet excited-state or S1) or phosphorescence (energy decay from the lowest spin-triplet excited-state or T1) processes, and has already found commercial applications in OLEDs and other envisioned uses. This mechanism is known as (1st generation) Thermally Activated Delayed Fluorescence (TADF) with the up-conversion of excitons relying necessarily on a sufficiently low S1-T1 energy difference, or simply ΔE_{ST} , together with a non-negligible spin-orbit coupling acting as the pre-factor of the corresponding RISC rate. The use of theoretical models has fostered the disclosure of helpful structure-property relationships to minimize the ΔE_{ST} value, as it happens e.g. for charge-transfer S1 and/or T1 excitations which thus reduces the frontier molecular orbital overlap and concomitantly the ΔE_{ST} value through a small exchange integral. On the other hand, that small overlap found led unfortunately to negligible (and thus in principle undesired) oscillator strength (f) values unless conformational (or dynamic) disorder had a predominant role.

However, the recent discovery of multi-resonant all-organic emitters (dubbed also as Hatakeyama's compounds) based on local heterosubstitution (C by N and/or B atoms) of aromatic compounds has shown the limitations of the (simplified) picture described above. For this case too, computational and theoretical studies have shown remarkably useful to analyze the physical origin of that mechanism, tangentially related with the content of the manuscript under evaluation. This (2nd generation) TADF mechanism has achieved considerably low ΔE_{ST} values for a large family of compounds, together with moderate oscillator strength values, which could pave the way towards more efficient OLEDs. Note also that the theoretical studies pioneeringly performed on these systems also identified the key role played by double-excitations to form the involved S1 and T1 excited-states, and thus the need to overcome the CIS/TD-DFT framework so widely used (before and now) within the field. Therefore, reconciling those two extremes (systems displaying sufficiently low ΔE_{ST} and moderate f values) from the materials design point of view was still pending until the discovery of those Hatakeyama's compounds.

Furthermore, the heptazine-based systems investigated here also combine the best of both worlds; i.e., molecules displaying small exchange integral values, for having low ΔE_{ST} values, and double-excitation effects, which could even invert the sign of ΔE_{ST} , while providing non-negligible oscillator strength values. Interestingly, the outcome of these efforts was also the discovery, or better said the

experimental confirmation, of the existence of systems displaying even negative ΔE_{ST} values, thus constituting a new (3rd) generation of TADF emitters what the authors rebaptised here as H-type delayed fluorescence. The lead candidate molecules disclosed after the screening of many compounds (HzTFEX2 and HzPipX2) are unsymmetrically substituted to reduce the symmetry point group (D3h) of pristine heptazines, which is known to be detrimental for emission: dark states are obtained driven by the symmetry of the transition dipole moment, a fact also indicated before in recent studies.

Given this historical perspective, and although the authors recognize and explain some of the previous efforts within the field, my first concern with the manuscript is the marginal reference to this path and, particularly, to some previous key findings leading to the current situation, i.e. the Hatekayama's compounds, from both experimental and computational points of view. Additionally, a recent study also screened a large number of compounds based on asymmetrical substitution of N-doped heptazine-like cores, which the authors only marginally mention as a reference. Going into the computational details, which is my area of expertise, I have the following remarks that should be carefully addressed by the authors. Note that any subtle (or considered as marginal, generally speaking) computational effect could significantly impact on such a low ΔE_{ST} value of -11 meV, which is why I insist so much on all these technical/computational issues.

(i) The manuscript specifies that the T1 geometry is optimized at the LC-BLYP/6-31G level, but should one assume that S0 is optimized at that level of theory too? Are thus the TD-DFT results for S1 and T1 excitation energies vertical or adiabatic? In any case, are all-real frequencies found for the lead candidate molecules HzTFEX2 and HzPipX2? Then, the authors mention that the T1 geometry was also optimized at the MP2/6-31G level, without further commenting if there are some differences with respect to LC-BLYP and/or the consequences of that. What I see as still more confusing is the footnote of Table S1, which explicitly mentions that the results reported there are calculated "at the lowest-energy triplet excited state (T1) optimized geometry". What does that statement mean? And why the S1 optimized geometry is excluded from that consideration?

(ii) The cost-efficient screening of such a large number of compounds needed of course to be done by a low-cost method, here resorting to LC-BLYP (authors are probably not aware of the sTD-DFT scheme) However, the range-separation parameter used (0.18 bohr⁻¹) is considerably different from the default value (0.40 bohr⁻¹) and some particular explanation is thus needed beyond a general statement like "to incorporate a reasonable amount of exact-exchange". Note that exact-exchange can be always incorporated by a global hybrid functional, but the range-separated LC-BLYP performance highly depends on that range-separation parameter. Additionally, the authors should specify if the Tamm-Dancoff approximation was imposed or not for the TD-DFT calculations, which is known to better describe the T1 excited states.

(iii) The EOM-CCSD, ADC(2), and CASPT2 methods are all applied with the cc-pVDZ basis set, and thus excluding diffuse functions. Could the authors do some test (e.g. aug-cc-pVDZ) to discard any influence of this extension on the results? Do the authors employ any numerical approximation (RI or any other density fitting technique) to alleviate the computational cost of these methods? If so, it should be specified as part of the computational details. Could the authors also consider ADC(3) or SCS-ADC(2) methods to better bracket the accuracy of the values found by ADC(2)? Finally, why

ADC(2) was chosen instead of CC2?

(iv) The CASPT2 method seems to largely overestimate the $\Delta\text{EST} < 0$ value, but I consider it could be a bit unfair to (even marginally) question the performance of that method without assessing the dependence of the results with respect to its many technicalities. Have the authors tried to systematically increase the active space, from e.g. (6,6) to (12,12) to see the trend followed by the results and especially ΔEST ? What was the criteria followed to select the active space? I guess the CASSCF underlying calculation are based on the HF orbitals, but the use of MP2 natural orbitals instead could facilitate the convergence of the calculations and possibly slightly modifying the excitation energies in the right direction. Is there any reason to prefer CASPT2 instead of NEVPT2?

(v) The use of any of the existing multi-configurational methods, as CASPT2, is expected to incorporate non-dynamical correlation effects in an efficient way. Of course, if the systems tackled are moderately radicaloid, or suffer from orbital near-degeneracies, the results from these methods might only agree with those ab initio methods including high-order excitations and not with those more severely truncated. As an indication of the radical-like nature of these HzTFEX2 and HzPipX2 systems, the authors might also report CASSCF natural occupation numbers and/or some other metrics as the NU values. Actually, the statement "double-excitation configurations with weights of approximately 1%" needs further clarification about how those weights are extracted and quantified.

(vi) The extended data (Table 1) includes HzTFEP2 and HzTFET2, a pair of closely related molecules to HzTFEX2 also showing negative ΔEST values, for which the authors could also perform (at least some of the) ab initio calculations and thus confirm the robustness of the whole computational protocol followed. The authors could also compare, for all the molecules experimentally explored, not only theoretical and experimental ΔEST values but also individual excitation energies to discard any compensation of errors on the final results.

Overall, the experimental confirmation of a negative ΔEST energy difference is undoubtedly a major breakthrough for the field, with the multi-step computational protocol the first step needed to clearly discard or identify a small but reasonable set of promising candidates. This is really a fascinating field, defying the conventional guidelines, and the contribution of the present manuscript could find definitively its place if an additional effort by the authors could better benchmark the outcome of the calculations.

Referee #2 (Remarks to the Author):

The authors report the observation of a negative energy gap between the lowest spin-singlet and spin-triplet state of a fluorescent molecule that they have synthesized, characterized by optical spectroscopy and calculated by TDDFT. The existence of molecules with such a negative exchange energy has been predicted, yet, if true, this would be the first experimental observation of this phenomena. It is a very important topic for two reasons. From a fundamental science point of view, it is experimental evidence that Hund's rule can be broken in certain circumstances. From an applied science view, it allows for the fabrication of OLEDs with 100% internal quantum efficiency without

use of heavy metals or thermally activated transfer from a triplet state. This goes beyond the current state of the art, as the conversion from an energetically higher triplet to a lower energy singlet should be fast (and this is also what the authors report), so that the lifetime of the excitations is short. This is expected to significantly reduce roll-off, which is a major problem in the development of OLEDs.

Thus, while the topic is very exciting and would warrant publication in nature, I am afraid I am not convinced by the spectroscopic evidence provided.

1. The main evidence for the negative singlet-triplet gap is based on fitting eq. 1 to the data, deriving the rates for k_{ISC} and k_{RISC} from this, plotting the coefficients for k_{ISC} and k_{RISC} as an Arrhenius plot, determining the activation energy for both rates, and subtracting them from each other, and taking the difference in the activation energies to be the difference in the singlet and triplet energy. Unfortunately, this procedure is not presented in a transparent way. I do not understand how eq. 1 was fitted to the data in Fig. 3b or Fig. 3c. Eq. 1 is actually a set of TWO equations, one for the decay of the singlet state, and one for the triplet state. Fig. b shows the PL decay, which the authors argue contains prompt and delayed fluorescence, and corresponds to a biexponential decay. Eq. 1 contains 4 rates (k_r , k_{nr} , k_{isc} , k_{risc}) just for the decay of the singlet, and 2 rates for the triplet decay. Which part of eq. 1 was fitted to which part of the curve, and how is it possible to disentangle the two rates k_{isc} and k_{risc} from this, given that k_r and k_{nr} are not known (since the lifetime contains all 4 rates)? The fits themselves are not shown, so it is not possible to assess their quality.

2. The observation that the PL quantum yields reduce upon aerating the solution to roughly 2/3 of their initial value is taken as evidence that triplet states are involved in creating delayed fluorescence. I am afraid, but this level of intensity reduction may arise from triplet quenching, yet it may also arise from photooxidation of the sample, involving only singlet state quenching.

3. The authors report the simultaneous decay of a signal at 700nm and a rise of a signal at 1600nm. They attribute this to S1-Sn absorption and T1-Tn absorption. What is supporting this assignment? Does the initial decay time, within the first 50ns, and the rise time of the T1-Tn match with the rates for ISC and RISC derived from the fitting of the PL decays in eq. 3? What happens if you aerate the solution? Do the transfer times remain, while the decay times shorten (perhaps even in agreement with the PL quantum yield reduction)? Moreover, do the initial transfer times slow down as you cool the solution (you may need to use MTHF if you want to go to very low temperatures)?

4. Does reduction in the lifetime of the delayed fluorescence from 217 to 195 ns when going from 300K to 200K match with what you would expect for a gap for -11 meV? (a simple back of the envelope calculation by myself seemed to indicate that a larger reduction should be expected) What lifetime do you get at very low temperatures such as below 100K, where any thermal activation to the higher lying triplet should be frozen out? Does the delayed fluorescence then vanish?

5. Why is the intensity drop between the initial prompt fluorescence and the delayed fluorescence so small? Would you not expect that the large majority of the singlets should decay with k_r , within few ns, and only a small fraction should be thermally activated to populate the triplet state and then drop back to the singlet?

6. The strong roll-off in the EL spectra is difficult to reconcile with a short lifetime in the triplet state.

7. There are a number of smaller issues:

- The Arrhenius plots for k_{ISC} and k_{RISC} would be better placed in either the manuscript, or the Supporting information, or as a clearly marked appendix to the manuscript before the references.

Just adding them at the end of the entire manuscript document is a little confusing.

- In Fig. 3a, it is not possible to see the S1 absorption. Perhaps you could add this part of the figure, e.g., magnified by a suitable factor, so that the reader can distinguish it from the baseline
- In Fig. 3b and 3c, one cannot see the prompt fluorescence. A display on a log-log scale might be more suitable here and is more common in the field of TADF emitters. Also, adding the biexponential fit to it would be useful. Further, I would find it more helpful if you gave the temperature in K, and not in inverted K. There is no reason here to quote temperature in inverted units.

Overall, I fear that in the present form, the conclusion is not sufficiently supported by the data, and I do not recommend publication at this stage. I feel the topic is very interesting, and the paper reads well, and a carefully revised version might merit consideration.

Referee #3 (Remarks to the Author):

The article by Aizawa, Pu, Miyajima et al presents some interest in the field of heptazines and TADF molecules, is correctly written, and presents reliable data. However, in my opinion it is far from presenting enough originality and novelty to warrant publication in a top general journal like nature. I precise my point of view hereunder

The discovery of the TADF emission of heptazines was made by C. Adachi in 2013 and 2014, who published almost simultaneously two papers (on symmetrical aryl heptazines). They also demonstrated, in one case with the same external emission yield (EQE) of 17%, the feasibility of a performing OLED.

A little later, W. Domcke published a series of 4 papers detailing the beautiful discovery, through a theoretical approach, of the fact that most heptazines should present themselves with the S1-T1 inversion.

Finally, the synthetic chemistry presented is nothing new, nucleophilic substitution on trichloroheptazine having been extensively documented by E. Kroke (and L. Dubois for amines), while the discovery of the pseudo-electrophilic substitution was discovered much earlier before by E. Kober in 1962. Similarly examples of unsymmetrical heptazines can be found here and there in the literature...

Therefore the only novelty of this article is the idea of screening heptazines using a coarse calculation method, which relies on the (very reasonable) that coarse calculations can be correlated through the entire family, to extract the "interesting" ones. However, the S1-T1 inversion, contrarywise to stated by the authors, although an amusing exotic feature confirming Domcke's calculations, is not the key point in this family of compounds...

There are other points requiring concern, the absence of which, also lowers the overall interest of the work.

There is first a ground problem with the authors' approach. It is true that they present the first example of an heptazine with a demonstrated T1-S1 inversion. Meanwhile, none of the (rare) previous authors interested in other TADF heptazines took this into consideration, probably because this is actually not a key point. Adachi did not put interest on the matter, neither P. Audebert, who were the first only groups to raise interest in the field (4 reports altogether). While with the fully aromatic conjugated symmetrical Adachi's heptazines, I would suspect the triplet to be slightly below the singlet, on the other hand I am convinced that if one would check the temperature-

dependance of the fluorescence of Audebert's alkoxy heptazines (Chem Commun, 2020) one would observe the same S1-T1 inversion, with at least a couple of them. This is because the reported decay times are quite in line with the ones reported here (by the way, the authors of the present report, although for an alleged different reason, also contains an heptazine also with one TFE substituent! A fact that would have certainly deserved to be noticed, and discussed a little...).

The reason for these "omissions" in previous reports is likely that, as far as TADF is pursued on, the fact that the triplet lies a little below, at the same level or a little above the singlet is actually unimportant, since extensive TADF will indeed always be observed at 300°K. On the other hand, and as Adachi realized and extensively discussed, the important point is the quantum yield. On this point, the two heptazines presented are interesting, while again not exceptional, with yields in the 50-70% range, lying in between Adachi's (80-90%) and Audebert's (20-30%) ones. Anyhow, despite the four Adachi's and Audebert's papers are correctly cited, their results must nevertheless be thoroughly discussed in a new version of this work, which has exactly the same points of interest.

The previous comment takes me to my last important point of concern. Unfortunately, no coarse theoretical approach (at least as far as I know) is able to, even grossly, predict the fluorescence yield of an heptazine, let alone another fluorophore family. But on the other hand, from the authors' data on Fig. 1d, I disagree on the view of a tradeoff between f and $\Delta E(ST)$. Meanwhile, looking closer at the data on Fig. 1d, it shows indeed a strong correlation (with very few exceptions!) between the main bandgap and the $\Delta E(ST)$. (said otherwise, most heptazines with (relatively...) split T and S are also blue emitting, while the ones with closer S and T are yellow to red emitting). This is quite interesting indeed, and overlooked in the paper; conversely the rare exceptions should deserve more attention. On the other hand, f seems to be much less influent (on $\Delta E(ST)$), while it might have huge consequences on the QY, as very frequently observed. This also would make an interesting addition to the article. As blue emitters are highly desirable, and high f values are often connected to high QYs, I would rather look for heptazines in the 0.2-0.3/0.1-0.15 zone of the Fig. 1d graph, where a few blue points are still present...

Minor point: On the four remarkable papers of Domcke on the heptazines' fluorescence, only one is cited, a situation which needs to be corrected when submitting to another journal; A recent 2021 Chem Rev by Kroke, Audebert et al., making the point on molecular heptazines, that the authors probably have missed, must of course also be cited, and its content might in addition help to write down the suggested discussions on the aforementioned points.

Author Rebuttals to Initial Comments:

Referee #1 (Remarks to the Author):

The interconversion of triplet to singlet excitons (formed upon a 3:1 ratio following the spin statistics) by a Reverse Intersystem Crossing Process (RISC) has received much attention as a promising mechanism to maximize the efficiency of light-emission, thus minimizing the loss of common fluorescence (energy decay from the lowest spin-singlet excited-state or S1) or phosphorescence (energy decay from the lowest spin-singlet excited-state or T1) processes, and has already found commercial applications in OLEDs and other envisioned uses. This mechanism is known as (1st generation) Thermally Activated Delayed Fluorescence (TADF) with the up-conversion of excitons relying necessarily on a sufficiently low S1-T1 energy difference, or simply ΔE_{ST} , together with a non-negligible spin-orbit coupling acting as the pre-factor of the corresponding RISC rate. The use of theoretical models has fostered the disclosure of helpful structure-property relationships to minimize the ΔE_{ST} value, as it happens e.g. for charge-transfer S1 and/or T1 excitations which thus reduces the frontier molecular orbital overlap and concomitantly the ΔE_{ST} value through a small exchange integral. On the other hand, that small overlap found led unfortunately to negligible (and thus in principle undesired) oscillator strength (f) values unless conformational (or dynamic) disorder had a predominant role.

However, the recent discovery of multi-resonant all-organic emitters (dubbed also as Hatakeyama's compounds) based on local heterosubstitution (C by N and/or B atoms) of aromatic compounds has shown the limitations of the (simplified) picture described above. For this case too, computational and theoretical studies have shown remarkably useful to analyze the physical origin of that mechanism, tangentially related with the content of the manuscript under evaluation. This (2nd generation) TADF mechanism has achieved considerably low ΔE_{ST} values for a large family of compounds, together with moderate oscillator strength values, which could pave the way towards more efficient OLEDs. Note also that the theoretical studies pioneeringly performed on these

systems also identified the key role played by double-excitations to form the involved S1 and T1 excited-states, and thus the need to overcome the CIS/TD-DFT framework so widely used (before and now) within the field. Therefore, reconciling those two extremes (systems displaying sufficiently low ΔE_{ST} and moderate f values) from the materials design point of view was still pending until the discovery of those Hatakayama's compounds. Furthermore, the heptazine-based systems investigated here also combine the best of both worlds; i.e., molecules displaying small exchange integral values, for having low ΔE_{ST} values, and double-excitation effects, which could even invert the sign of ΔE_{ST} , while providing non-negligible oscillator strength values. Interestingly, the outcome of these efforts was also the discovery, or better said the experimental confirmation, of the existence of systems displaying even negative ΔE_{ST} values, thus constituting a new (3rd) generation of TADF emitters what the authors rebaptised here as H-type delayed fluorescence. The lead candidate molecules disclosed after the screening of many compounds (HzTFEX2 and HzPipX2) are unsymmetrically substituted to reduce the symmetry point group (D_{3h}) of pristine heptazines, which is known to be detrimental for emission: dark states are obtained driven by the symmetry of the transition dipole moment, a fact also indicated before in recent studies.

Given this historical perspective, and although the authors recognize and explain some of the previous efforts within the field, my first concern with the manuscript is the marginal reference to this path and, particularly, to some previous key findings leading to the current situation, i.e. the Hatakayama's compounds, from both experimental and computational points of view. Additionally, a recent study also screened a large number of compounds based on asymmetrical substitution of N-doped heptazine-like cores, which the authors only marginally mention as a reference. Going into the computational details, which is my area of expertise, I have the following remarks that should be carefully addressed by the authors. Note that any subtle (or considered as marginal, generally speaking) computational effect could significantly impact on such a low ΔE_{ST} value of -11 meV, which is why I insist so much on all these technical/computational issues.

Re: We have thankfully highlighted the two key findings in the revised manuscript as follows:

“We note that the accounting for double-excitation configurations has proven crucial to theoretically reproduce the small but positive ΔE_{ST} of 5,9-diphenyl-5,9-diaza-13b-boranaphtho[3,2,1-*de*]anthracene (DABNA-1) (0.15 eV)^{22,23}.” on page 2, line 58.

“Furthermore, the recent computational screening by Pollice et al. has demonstrated that appropriate chemical modifications of heptazine recover f while retaining negative ΔE_{ST} ¹⁹.” on page 2, line 65.

“We note that these molecules recover f while retaining small ΔE_{ST} ($f = 0.010, 0.015$ and $\Delta E_{ST} = 210, 334$ meV for HzTFEX₂ and HzPipX₂, respectively). This trend is consistent with the recent computational screening on heptazine analogues with asymmetrical substitutions¹⁹.” on page 3, line 94.

“Similar spatial separations of NTOs have also been found in the multi-resonant TADF materials, such as DABNA-1^{22,23}.” on page 4, line 110.

We also appreciate the valuable remarks on the calculations. As explained in the below point-by-point response, we have revised manuscript by including the details of the calculations and the additional calculation results.

(i) The manuscript specifies that the T₁ geometry is optimized at the LC-BLYP/6-31G level, but should one assume that S₀ is optimized at that level of theory too? Are thus the TD-DFT results for S₁ and T₁ excitation energies vertical or adiabatic?

Re: No. The geometry is optimized for T₁ and not for S₀ at the LC-BLYP/6-31G level. Thus, the TD-DFT results for S₁ and T₁ excitation energies are vertical. This information can be found on page 11, line 346:

“The vertical excitation energies of S₁ and T₁ were calculated using linear-response TDDFT with the LC-BLYP functional and the 6-31G(d) basis set.”

To improve the clarity of this point, we have added “vertical” on page 3, line 91.

In any case, are all-real frequencies found for the lead candidate molecules HzTFEX₂ and HzPipX₂?

Re: Yes. To specify this, we have added the following sentence on page 11, line 345:

“Vibrational frequency analysis for HzTFEX₂, HzPipX₂, HzTFET₂, and HzTFEP₂ gave no imaginary frequencies at the same level of theory.”

Then, the authors mention that the T₁ geometry was also optimized at the MP2/6-31G level, without further commenting if there are some differences with respect to LC-BLYP and/or the consequences of that.

Re: The basis set used with MP2 is cc-pVDZ and not 6-31G. The computational cost of MP2/cc-pVDZ is much higher than that of DFT(LC-BLYP)/6-31G. Thus, MP2/cc-pVDZ is not practical for the screening of the thirty thousand molecules. MP2/cc-pVDZ was also employed to optimize the geometry in the prior theoretical studies on heptazine (Ref. 15).

What I see as still more confusing is the footnote of Table S1, which explicitly mentions that the results reported there are calculated “at the lowest-energy triplet excited state (T₁) optimized geometry”. What does that statement mean?

Re: We meant that the results were calculated using the geometry optimized for T_1 . To improve the clarity of this point, we have modified the foot note of Supplementary Table 1 to:

“^aTDDFT LC-BLYP/6-31G(d) using the geometry optimized for the lowest-energy triplet excited state (T_1) by unrestricted LC-BLYP/6-31G with the range-separated parameter of 0.18 bohr⁻¹. ^bEOM-CCSD/cc-pVDZ using the T_1 geometry optimized by unrestricted MP2/cc-pVDZ. ^cADC(2)/cc-pVDZ using the T_1 geometry optimized by unrestricted MP2/cc-pVDZ. ^dCASPT2(12,12)/cc-pVDZ using the T_1 geometry optimized by unrestricted MP2/cc-pVDZ.”

And why the S_1 optimized geometry is excluded from that consideration?

Re: Because unrestricted MP2 is not able to give the S_1 optimized geometry. EOM-CCSD, ADC(2), and CASPT2 afford geometry optimization of both S_1 and T_1 , but their gradient calculations require unrealistic computational cost for a relatively large molecule like HzTFEX₂. The S_1 optimized geometry was also excluded from the consideration in the prior theoretical studies on heptazine and cycl[3.3.3]azine (ref. 15 and 16).

(ii) The cost-efficient screening of such a large number of compounds needed of course to be done by a low-cost method, here resorting to LC-BLYP (authors are probably not aware of the sTD-DFT scheme) However, the range-separation parameter used (0.18 bohr⁻¹) is considerably different from the default value (0.40 bohr⁻¹) and some particular explanation is thus needed beyond a general statement like “to incorporate a reasonable amount of exact-exchange”. Note that exact-exchange can be always incorporated by a global hybrid functional, but the range-separated LC-BLYP performance highly depends on that range-separation parameter.

Re: To specify this point, we have modified the explanation on page 11, line 348 to:

“The range-separation parameter of the LC-BLYP functional was non-empirically optimised for a heptazine analogue with three phenyl groups, 2,5,8-triphenylheptazine, to 0.18 bohr⁻¹ to reasonably weight the Hartree–Fock (HF) exchange and the DFT exchange⁴⁷.”

Additionally, the authors should specify if the Tamm-Dancoff approximation was imposed or not for the TD-DFT calculations, which is known to better describe the T_1 excited states.

Re: We used the Tamm–Dancoff approximation for the TD-DFT calculations. To specify this, we have added “within the Tamm–Dancoff approximation” on page 11, line 348.

(iii) The EOM-CCSD, ADC(2), and CASPT2 methods are all applied with the cc-pVDZ basis set, and thus excluding diffuse functions. Could the authors do some test (e.g. aug-cc-pVDZ) to discard any influence of this extension on the results?

Re: We have calculated ΔE_{ST} of heptazine using various methods with the aug-cc-pVDZ basis set and listed the results in Supplementary Table 8. The ΔE_{ST} values appear not to be so sensitive to the extension of the basis set from cc-pVDZ to aug-cc-pVDZ.

Supplementary Table 8 | Vertical S_1 and T_1 excitation energies and ΔE_{ST} of heptazine, calculated by various methods with the cc-pVDZ and aug-cc-pVDZ basis sets.

Method ^a	Basis	S_1 excitation energy (eV)	T_1 excitation energy (eV)	ΔE_{ST} (meV)
EOM-CCSD	cc-pVDZ	2.656	2.838	-182
	aug-cc-pVDZ	2.727	2.884	-157
LT-DF-LCC2	cc-pVDZ	2.552	2.817	-265
	aug-cc-pVDZ	2.572	2.831	-259
ADC(2)	cc-pVDZ	2.438	2.723	-285
	aug-cc-pVDZ	2.474	2.743	-269
SCS-ADC(2)	cc-pVDZ	2.364	2.895	-531
	aug-cc-pVDZ	2.428	2.926	-498
CASPT2(12,12)	cc-pVDZ	2.302	2.511	-209
	aug-cc-pVDZ	2.259	2.395	-136

^aAll of the calculations used the T_1 geometry optimized by unrestricted MP2/cc-pVDZ.

Do the authors employ any numerical approximation (RI or any other density fitting technique) to alleviate the computational cost of these methods? If so, it should be specified as part of the computational details.

Re: We used the resolution of identity (RI) approximation in the CASPT2 calculations. To specify this point, we have added “using the resolution of identity approximation with the auxiliary fitting basis set.” on page 11, line 356.

Could the authors also consider ADC(3) or SCS-ADC(2) methods to better bracket the accuracy of the values found by ADC(2)? Finally, why ADC(2) was chosen instead of CC2?

Re: We have performed additional calculations of ΔE_{ST} of HzTFEX₂ and HzPipX₂ using the SCS-ADC(2) and listed the results in Supplementary Table 6. Both ADC(2) and CC2 are second-order methods for calculating excitation energies and could be suitable to simulate the inversion of S_1 and T_1 of the heptazine analogues. Initially we did not have access to a program implementing any CC2

methods. Stimulated by the question from Referee#1, we have performed local coupled cluster (LCC2) calculations using a newly purchased MOLPRO program and listed the results in Supplementary Table 6. The LCC2 calculations used the Laplace transform (LT) and density fitting (DF) to reduce the computational costs.

Supplementary Table 6 | Vertical S_1 and T_1 excitation energies and ΔE_{ST} of HzTFEX₂ and HzPipX₂, calculated by LT-DF-LCC2/cc-pVDZ and SCS-ADC(2)/cc-pVDZ.

Molecule	Method	S_1 excitation energy (eV)	T_1 excitation energy (eV)	ΔE_{ST} (meV)
HzTFEX ₂	LT-DF-LCC2 ^a	2.319	2.347	-28
	SCS-ADC(2) ^b	2.301	2.447	-146
HzPipX ₂	LT-DF-LCC2 ^a	2.715	2.735	-20
	SCS-ADC(2) ^b	2.635	2.753	-118

^aLT-DF-LCC2/cc-pVDZ using the T_1 geometry optimized by unrestricted MP2/cc-pVDZ. ^bSCS-ADC(2)/cc-pVDZ using the T_1 geometry optimized by unrestricted MP2/cc-pVDZ. The LT-DF-LCC2 calculations were performed using the MOLPRO 2019.2 program⁶.

(iv) The CASPT2 method seems to largely overestimate the $\Delta E_{ST} < 0$ value, but I consider it could be a bit unfair to (even marginally) question the performance of that method without assessing the dependence of the results with respect to its many technicalities. Have the authors tried to systematically increase the active space, from e.g. (6,6) to (12,12) to see the trend followed by the results and especially ΔE_{ST} ?

Re: Yes. We have included the CASPT2 results with different active spaces in Supplementary Table 4. The trend of the overestimated $\Delta E_{ST} < 0$ were still observed. The larger active spaces, (10, 10) and (12,12), reproduced the trend of ΔE_{ST} of HzTFEX₂ < ΔE_{ST} of HzPipX₂. Please note that the CASPT2 method in this study was not used to accurately reproduce the experimental ΔE_{ST} , but was used to examine the possibilities of the molecules to have negative ΔE_{ST} prior to the experimental evaluation. Thus, the manuscript does not question the performance of the CASPT2 method, but rather pointing out the variation in ΔE_{ST} across the calculation methods and the importance of experimental evaluation.

Supplementary Table 4 | Active-space dependence of the vertical S_1 and T_1 excitation energies and ΔE_{ST} of HzTFEX₂ and HzPipX₂, calculated by CASPT2/cc-pVDZ using CASSCF based on HF orbitals.

Molecule	Active space	S_1 excitation energy (eV)	T_1 excitation energy (eV)	ΔE_{ST} (meV)
HzTFEX ₂	(6,6)	2.213	2.271	-58
	(8,8)	1.719	1.759	-40
	(10,10)	1.558	1.811	-253
	(12,12)	2.037	2.221	-184
HzPipX ₂	(6,6)	1.713	3.102	-1389
	(8,8)	2.557	2.711	-154
	(10,10)	2.393	2.509	-116
	(12,12)	2.496	2.325	-171

What was the criteria followed to select the active space?

Re: Generally speaking, the active space should be enough large to correctly describe the electronic structure for accurate CASSCF/CASPT2 calculations, though the computational cost exponentially increases with extending the active space. We selected (12,12) because it is the largest active space for practical calculations prior to the synthesis of a relatively large molecule like HzTFEX₂.

I guess the CASSCF underlying calculation are based on the HF orbitals, but the use of MP2 natural orbitals instead could facilitate the convergence of the calculations and possibly slightly modifying the excitation energies in the right direction.

Re: We have performed additional calculations using MP2 natural orbitals for the CASSCF and listed the CASPT2 results in Supplementary Table 5. The calculations predicted negative ΔE_{ST} as similar to those based on the HF orbitals.

Supplementary Table 5 | Active-space dependence of the vertical S_1 and T_1 excitation energies and ΔE_{ST} of HzTFEX₂ and HzPipX₂, calculated by CASPT2/cc-pVDZ using CASSCF based on MP2 natural orbitals.

Molecule	Active space	S_1 excitation energy (eV)	T_1 excitation energy (eV)	ΔE_{ST} (meV)
HzTFEX ₂	(6,6)	2.749	2.861	-112
	(8,8)	1.592	2.736	-1207
	(10,10)	2.283	2.371	-88
	(12,12)	1.984	2.155	-171
HzPipX ₂	(6,6)	2.822	3.048	-226
	(8,8)	2.354	2.713	-359
	(10,10)	2.528	2.664	-136
	(12,12)	2.423	2.701	-278

Is there any reason to prefer CASPT2 instead of NEVPT2?

Re: CASPT2 and NEVPT2 are perturbative methods to similarly recover the dynamic correlation in CASSCF, though they use different definitions of the zeroth-order Hamiltonian. We used CASPT2 since it was used in the prior theoretical study on heptazine (Ref. 15).

(v) The use of any of the existing multi-configurational methods, as CASPT2, is expected to incorporate non-dynamical correlation effects in an efficient way. Of course, if the systems tackled are moderately radicaloid, or suffer from orbital near-degeneracies, the results from these methods might only agree with those ab initio methods including high-order excitations and not with those more severely truncated. As an indication of the radical-like nature of these HzTFEX₂ and HzPipX₂ systems, the authors might also report CASSCF natural occupation numbers and/or some other metrics as the NU values.

Re: The CASSCF natural occupation numbers of HOMO/LUMO of HzTFEX₂ and HzPipX₂ are 1.2496/0.7600 and 1.2586/0.7459, respectively. However, these values do not necessarily indicate the radical-like nature of the molecules in the ground states (S_0) since the calculations were based on the state-averaged CASSCF for S_0 , S_1 , and T_1 (Methods on page 11, line 355). Instead, we have included the compositions of the CASSCF wave functions for each state in Supplementary Table 9. The data indicate that the S_0 of the four synthesized molecules are almost described by the closed-shell configuration. Additionally, the S_1 of the molecules are dominated by the single-excitation configuration with the small contributions from the multiple-excitation configuration, which are slightly higher than those in the T_1 .

Supplementary Table 9 | The compositions of the wave functions in the CASSCF(12,12)/cc-pVDZ calculations.

Molecule	State	Configuration	Weight (%) ^a
HzTFEX ₂	S ₀	Closed shell	86.9
		Single excitation	4.1
		Multiple excitation	5.3
	S ₁	Closed shell	3.0
		Single excitation	86.0
		Multiple excitation	8.5
	T ₁	Closed shell	–
		Single excitation	89.7
		Multiple excitation	7.5
HzPipX ₂	S ₀	Closed shell	83.4
		Single excitation	3.4
		Multiple excitation	5.0
	S ₁	Closed shell	2.8
		Single excitation	79.1
		Multiple excitation	8.9
	T ₁	Closed shell	–
		Single excitation	83.9
		Multiple excitation	8.0
HzTFEP ₂	S ₀	Closed shell	81.8
		Single excitation	3.7
		Multiple excitation	2.8
	S ₁	Closed shell	1.4
		Single excitation	77.5
		Multiple excitation	8.8
	T ₁	Closed shell	–
		Single excitation	83.2
		Multiple excitation	6.3
HzTFET ₂	S ₀	Closed shell	81.8
		Single excitation	3.9
		Multiple excitation	2.5
	S ₁	Closed shell	1.5
		Single excitation	77.2
		Multiple excitation	7.6
	T ₁	Closed shell	–
		Single excitation	83.2
		Multiple excitation	4.9

^aSum of the configuration weights of > 0.25%.

Actually, the statement “double-excitation configurations with weights of approximately 1%” needs further clarification about how those weights are extracted and quantified.

Re: To specify this point, we have modified the sentence on page 4, line 112 to:

“Indeed, S_1 of both molecules comprise double-excitations configurations with weights of around 1% described as the sum of the squares of the doubles amplitudes in EOM-CCSD, which are slightly higher than those of T_1 .”

(vi) The extended data (Table 1) includes HzTFEP2 and HzTFET2, a pair of closely related molecules to HzTFEX2 also showing negative ΔE_{ST} values, for which the authors could also perform (at least some of the) ab initio calculations and thus confirm the robustness of the whole computational protocol followed.

Re: We have performed additional calculations for HzTFEP₂ and HzTFET₂ and listed the results in Supplementary Table 7. All of the methods except for TDDFT predicted negative ΔE_{ST} for both molecules.

Supplementary Table 7 | Vertical S₁ and T₁ excitation energies, ΔE_{ST}, and *f* of HzTFEP₂ and HzTFET₂, calculated by various methods.

Molecule	Method	S ₁ excitation energy (eV)	T ₁ excitation energy (eV)	ΔE _{ST} (meV)	f
HzTFEP ₂	TDDFT ^a	2.729	2.464	265	0.008
	EOM-CCSD ^b	2.892	2.925	-33	0.013
	LT-DF-LCC2 ^c	2.650	2.765	-115	-
	ADC(2) ^d	2.156	2.197	-41	0.019
	SCS-ADC(2) ^e	2.545	2.878	-333	0.015
	CASPT2(12,12) ^f	2.108	2.463	-355	0.167
HzTFET ₂	TDDFT ^a	2.748	2.489	259	0.009
	EOM-CCSD ^b	2.899	2.944	-45	0.011
	LT-DF-LCC2 ^c	2.696	2.786	-90	-
	ADC(2) ^d	2.147	2.229	-82	0.025
	SCS-ADC(2) ^e	2.601	2.867	-266	0.014
	CASPT2(12,12) ^f	2.192	2.456	-264	0.198

^aTDDFT LC-BLYP/6-31G(d) using the geometry optimized for T₁ by unrestricted LC-BLYP/6-31G with the range-separated parameter of 0.18 bohr⁻¹. ^bEOM-CCSD/cc-pVDZ using the T₁ geometry optimized by unrestricted MP2/cc-pVDZ. ^cLT-DF-LCC2/cc-pVDZ using the T₁ geometry optimized by unrestricted MP2/cc-pVDZ. ^dADC(2)/cc-pVDZ using the T₁ geometry optimized by unrestricted MP2/cc-pVDZ. ^eSCS-ADC(2)/cc-pVDZ using the T₁ geometry optimized by unrestricted MP2/cc-pVDZ. ^fCASPT2(12,12)/cc-pVDZ using the T₁ geometry optimized by unrestricted MP2/cc-pVDZ.

The authors could also compare, for all the molecules experimentally explored, not only theoretical and experimental ΔEST values but also individual excitation energies to discard any compensation of errors on the final results.

Re: We have included the comparison of the theoretical and experimental excitation energies and ΔE_{ST} for all the molecules experimentally explored in Supplementary Table 10.

Supplementary Table 10 | Summary of theoretical and experimental excitation energies and ΔE_{ST} of HzTFEX₂, HzPipX₂, HzTFEP₂, and HzTFET₂.

Molecule	Method	S ₁ excitation energy (eV)	T ₁ excitation energy (eV)	ΔE_{ST} (meV)
HzTFEX ₂	TDDFT	2.708	2.498	210
	EOM-CCSD	2.678	2.690	-12
	LT-DF-LCC2	2.319	2.347	-28
	ADC(2)	2.199	2.233	-34
	SCS-ADC(2)	2.301	2.447	-146
	CASPT2(12,12)	2.037	2.221	-184
	Experiment ^a	2.76	2.77	-11
HzPipX ₂	TDDFT	2.840	2.506	334
	EOM-CCSD	3.032	3.022	10
	LT-DF-LCC2	2.715	2.735	-20
	ADC(2)	2.612	2.624	-12
	SCS-ADC(2)	2.635	2.753	-118
	CASPT2(12,12)	2.496	2.325	-171
	Experiment ^a	2.81	2.76	52
HzTFEP ₂	TDDFT	2.729	2.464	265
	LT-DF-LCC2	2.650	2.765	-115
	ADC(2)	2.156	2.197	-41
	SCS-ADC(2)	2.545	2.878	-333
	CASPT2(12,12)	2.108	2.463	-355
	Experiment ^a	2.73	2.72	-14
	HzTFET ₂	TDDFT	2.748	2.489
LT-DF-LCC2		2.696	2.786	-90
ADC(2)		2.147	2.229	-82
SCS-ADC(2)		2.601	2.867	-266
CASPT2(12,12)		2.192	2.456	-264
Experiment ^a		2.75	2.74	-13

^aExperimental S₁ excitation energy was estimated from the lowest-energy peak of the fluorescence spectra.

ΔE_{ST} was obtained as the difference between the activation energies of ISC and RISC. The T₁ excitation energy was estimated by subtracting ΔE_{ST} from the S₁ excitation energy.

Overall, the experimental confirmation of a negative ΔE_{ST} energy difference is undoubtedly a major breakthrough for the field, with the multi-step computational protocol the first step needed to clearly discard or identify a small but reasonable set of promising candidates. This is really a fascinating field, defying the conventional guidelines, and the contribution of the present manuscript could find definitively its place if an additional effort by the authors could better benchmark the outcome of the calculations.

Re: We thank Referee#1 for recognizing the novelty and impact of the work. We also appreciate the valuable suggestions for improving our manuscript.

Referee #2 (Remarks to the Author):

The authors report the observation of a negative energy gap between the lowest spin-singlet and spin-triplet state of a fluorescent molecule that they have synthesized, characterized by optical spectroscopy and calculated by TDDFT. The existence of molecules with such a negative exchange energy has been predicted, yet, if true, this would be the first experimental observation of this phenomena. It is a very important topic for two reasons. From a fundamental science point of view, it is experimental evidence that Hund's rule can be broken in certain circumstances. From an applied science view, it allows for the fabrication of OLEDs with 100% internal quantum efficiency without use of heavy metals or thermally activated transfer from a triplet state. This goes beyond the current state of the art, as the conversion from an energetically higher triplet to a lower energy singlet should be fast (and this is also what the authors report), so that the lifetime of the excitations is short. This is expected to significantly reduce roll-off, which is a major problem in the development of OLEDs. Thus, while the topic is very exciting and would warrant publication in nature, I am afraid I am not convinced by the spectroscopic evidence provided.

Re: We thank Referee#2 for remarking the novelty and impact of the work in both fundamental and applied sciences. As explained in the below point-by-point response, we have revised manuscript by including the details of the spectroscopic data.

1. The main evidence for the negative singlet-triplet gap is based on fitting eq. 1 to the data, deriving the rates for k_{ISC} and k_{RISC} from this, plotting the coefficients for k_{ISC} and k_{RISC} as an Arrhenius plot, determining the activation energy for both rates, and subtracting them from each other, and taking the difference in the activation energies to be the difference in the singlet and triplet energy. Unfortunately, this procedure is not presented in a transparent way. I do not understand how eq. 1 was fitted to the data in Fig. 3b or Fig. 3c. Eq. 1 is actually a set of TWO equations, one for the decay of the singlet state, and one for the triplet state. Fig. b shows the PL decay, which the authors argue contains prompt and delayed fluorescence, and corresponds to a biexponential decay. Eq. 1

contains 4 rates (k_r , k_{nr} , k_{isc} , k_{risc}) just for the decay of the singlet, and 2 rates for the triplet decay. Which part of eq. 1 was fitted to which part of the curve, and how is it possible to disentangle the two rates k_{isc} and k_{risc} from this, given that k_r and k_{nr} are not known (since the lifetime contains all 4 rates)? The fits themselves are not shown, so it is not possible to assess their quality.

Re: Since the time derivative of S_1 depends on the T_1 population and the time derivative of T_1 depends on the S_1 population, the rate equation Eq (1) must be this form of simultaneous linear differential equations. The S_1 population in Eq (1) was fitted to the transient PL decay in Fig 3b and 3c. To clarify this point, we modified explanation on page 12, line 381 to:

“ $k_r + k_{nr}$, k_{ISC} , and k_{RISC} were determined without assuming $k_{ISC} \gg k_{RISC}$ by fitting the S_1 population in Eq. (1) to the transient PL decay data”

The fitting gave 3 values, $k_r + k_{nr}$, k_{ISC} , and k_{RISC} . Then, k_r and k_{nr} were calculated using $\Phi_{PL} = k_r/(k_r + k_{nr})$. We have included the fitting curve in Extended Data Fig. 2. The fitting parameters well reproduce the transient PL decays of HzTFEX₂ and HzPipX₂ (with the determination coefficients $R^2 > 0.999$).

Extended Data Fig. 2 | Analysis of the transient PL decays of HzTFEX₂ and HzPipX₂. **a, b**, Fit (red solid line) of the S_1 population in Eq. (1) to the transient PL decay of HzTFEX₂ (a) and HzPipX₂ (b) at 300 K. The blue solid line represents the T_1 population simulated by the best-fit parameters $k_r + k_{nr}$, k_{ISC} , and k_{RISC} in Eq. (1).

Additionally, the T_1 population is lower than the S_1 population in HzTFEX₂, further ensuring the S_1 lies below the T_1 (Extended Data Fig. 2a). We note that the T_1 population of HzPipX₂ is higher than the S_1 population under the steady-state condition, reflecting the positive ΔE_{ST} (Extended Data Fig. 2b). To specify this point, we have added the following sentence on page 6, line 182:

“These parameters simulate that the population of T_1 is lower than that of S_1 in HzTFEX₂ under the steady-state condition, indicating that S_1 lies energetically below T_1 (Extended Data Fig. 2).”

2. The observation that the PL quantum yields reduce upon aerating the solution to roughly 2/3 of their initial value is taken as evidence that triplet states are involved in creating delayed fluorescence. I am afraid, but this level of intensity reduction may arise from triplet quenching, yet it may also arise from photooxidation of the sample, involving only singlet state quenching.

Re: The change in the PL quantum yield is reversible, excluding the possibility that the photooxidation of the sample. Thus, it is more reasonably attributed to the triplet quenching by atmospheric O₂. To specify this point, we have added “and the change in Φ_{PL} is reversible” in page 4 line 134.

3. The authors report the simultaneous decay of a signal at 700nm and a rise of a signal at 1600nm. They attribute this to S₁-S_n absorption and T₁-T_n absorption. What is supporting this assignment?

Re: As shown in Extended Data Fig. 1b, the assignment is supported by the fact that the reduction of the transient absorption upon aerating the solution is significant at 1600 nm, but not at 700 nm. We also note that the green dashed line in the Figure represents the spectral difference of the transient absorption of the deaerated and aerated solutions and mainly corresponds to the transient absorption of T₁.

Does the initial decay time, within the first 50ns, and the rise time of the T₁-T_n match with the rates for ISC and RISC derived from the fitting of the PL decays in eq. 3?

Re: We suppose that Referee#2 has meant Eq. 1. The initial decay time constant of the S₁-S_n is 27 ns corresponding to the rate constant of $3.7 \times 10^7 \text{ s}^{-1}$ and the rise time constant of the T₁-T_n is 45 ns correspond to the rate constant of $2.2 \times 10^7 \text{ s}^{-1}$. These rate constants are of the same order of magnitude as k_{ISC} ($2.3 \times 10^7 \text{ s}^{-1}$) and k_{RISC} ($4.2 \times 10^7 \text{ s}^{-1}$), though they should not be exactly same given the complex excited-state kinetics as described by Eq. 1.

What happens if you aerate the solution? Do the transfer times remain, while the decay times shorten (perhaps even in agreement with the PL quantum yield reduction)?

Re: Yes, the decay times of the transient absorption of S₁ and T₁ shorten upon aerating the solution. We have included the data in Supplementally Fig. 8. The reduction of the S₁ transient absorption upon aerating the solution is 21%, while 27% is observed for the PL quantum yield.

Supplementary Fig. 8 | Transient absorption of HzTFEX₂ in deaerated and aerated toluene solutions. Transient absorption decays of S₁ and T₁ monitored at 700 nm and 1600 nm, respectively, in deaerated and aerated toluene solutions.

Moreover, do the initial transfer times slow down as you cool the solution (you may need to use MTHF if you want to go to very low temperatures)?

Re: Yes, the initial decay and rise times slowed down at a low temperature. We have included the data in Supplementally Fig. 9.

Supplementary Fig. 9 | Transient absorption decay of HzTFEX₂ in a deaerated toluene solution at 300 K and 200 K. Transient absorption decays of S₁ and T₁ monitored at 700 nm and 1600 nm, respectively, in a deaerated toluene solution at 300 K and 200 K.

4. Does reduction in the lifetime of the delayed fluorescence from 217 to 195 ns when going from 300K to 200K match with what you would expect for a gap for -11 meV? (a simple back of the envelope calculation by myself seemed to indicate that a larger reduction should be expected)

Re: As Referee#2 pointed out, a simple Arrhenius-type equation $k_{DF} = A \exp(-\Delta E_{ST}/k_B T)$, where k_{DF} is the inverse of delayed fluorescence lifetime (217 ns–195 ns), gives ΔE_{ST} of -4 meV. However, this

approach neglects the temperature dependence of k_r and k_{nr} (generally k_{nr} is suppressed at low temperatures, increasing the excited-state lifetime). Thus, ΔE_{ST} of -11 meV obtained as the difference between the activation energies of ISC and RISC should be more accurate.

What lifetime do you get at very low temperatures such as below 100K, where any thermal activation to the higher lying triplet should be frozen out? Does the delayed fluorescence then vanish?

Re: To answer this question, we measured the transient PL decay of HzTFEX₂ in 2-methyltetrahydrofuran (MeTHF) at 80 K (the lowest temperature reached in our setup using liquid N₂). As shown in Fig. R1, the delayed fluorescence was still observed, which is natural given the small ΔE_{ST} close to the thermal energy at 80 K (7 meV). Please also note that MeTHF is indeed a good solvent for low-temperature spectroscopic measurements because it makes a good optical glass when frozen. However, its dielectric constant has a strong temperature dependence (7.0 at 300 K, 18.5 at 120 K, and 2.5 at 80 K). This means that lowering the temperature could change ΔE_{ST} from that at the room temperature in MeTHF (This is why MeTHF is not commonly used in TADF studies).

Fig. R1 | Transient PL decay of HzTFEX₂ in a deaerated MeTHF solution at 80 K.

5. Why is the intensity drop between the initial prompt fluorescence and the delayed fluorescence so small? Would you not expect that the large majority of the singlets should decay with k_r , within few ns, and only a small fraction should be thermally activated to populate the triplet state and then drop back to the singlet?

Re: No, the large majority of excitations decay to emit delayed fluorescence in HzTFEX₂ since k_{ISC} (2.3×10^7 s⁻¹) is larger than k_r (5.4×10^6 s⁻¹). These rate constants can be found in Fig. 3 e and Extended Data Table 1. We note that the PL decay of HzTFEX₂ in the initial 100 ns is not steep because the delayed fluorescence is so fast that both prompt and delayed fluorescence are mixed even in this initial time range. So too does in HzTFEP₂ and HzTFET₂. To improve the clarity of this

point, we have added the following Supplementary Fig. 6, Supplementary Fig. 7, and Supplementary Table 3:

Supplementary Fig. 7 | Biexponential fits of the transient PL decay. **a, b, c, d**, Transient PL decays of HzTFEX₂ (a), HzPipX₂ (b), HzTFEP₂ (c), and HzTFET₂ (d) in deaerated toluene solutions at 300 K. The red solid lines represent biexponential fits of the transient PL decays. The blue and green solid lines represent the components of the prompt fluorescence and the delayed fluorescence, respectively.

Supplementary Fig. 8 | Log-log representation of the biexponential fits of the transient PL decay. a, b, c, d, Transient PL decays of HzTFEX₂ (a), HzPipX₂ (b), HzTFEP₂ (c), and HzTFET₂ (d) in deaerated toluene solutions at 300 K. The red solid lines represent biexponential fits of the transient PL decays. The blue and green solid lines represent the components of the prompt fluorescence and the delayed fluorescence, respectively.

Supplementary Table 5 | Parameters of the biexponential fits of the transient PL decays.

Emitter	τ_{PF} (ns)	τ_{DF} (ns)	A_1	A_2	Φ_{PL} (%) ^a	Φ_{PF} (%) ^b	Φ_{DF} (%) ^c
HzTFEX ₂	14	217	0.45	0.55	74	8	66
HzPipX ₂	7.9	565	0.87	0.13	67	7	60
HzTFEP ₂	35	288	0.54	0.46	44	10	34
HzTFET ₂	23	246	0.44	0.56	42	7	35

^aPL quantum yield. ^bComponent of prompt fluorescence to Φ_{PL} . ^cComponent of prompt fluorescence to Φ_{PL} .

6. The strong roll-off in the EL spectra is difficult to reconcile with a short lifetime in the triplet state.

Re: We suppose that Referee#2 has meant the roll-off in the external quantum efficiency–luminance characteristics (Fig. 4b). This is most likely due to the large hole-injection barrier to the emission layer, causing the carrier imbalance at higher luminances as explained on page 8, line 223:

“Although the efficiency roll-off is still significant in this preliminary device concerning the large hole-injection barrier caused by the high ionization potential of HzTFEX₂ (6.3 eV), we anticipate that further optimisation of molecular design will address this issue and allow a conclusive exploration of the effects of negative ΔE_{ST} on efficiency roll-off and device stability.”

7. There are a number of smaller issues:

- The Arrhenius plots for k_{ISC} and K_{RISC} would be better placed in either the manuscript, or the Supporting information, or as a clearly marked appendix to the manuscript before the references. Just adding them at the end of the entire manuscript document is a little confusing.

Re: We have moved the Arrhenius plots for k_{ISC} and k_{RISC} (Extended Data Fig. 3) in the manuscript on page 7, line 196.

- In Fig. 3a, it is not possible to see the S1 absorption. Perhaps you could add this part of the figure, e.g., magnified by a suitable factor, so that the reader can distinguish it from the baseline

Re: We have added the magnified view of the steady-state absorption spectra to Fig. 3a. The modified Fig. 3 now appears as:

Fig. 3 | Photophysical properties of HzTFEX₂ and HzPipX₂ in deaerated toluene solutions. **a**, Steady-state absorption and PL spectra of HzTFEX₂ and HzPipX₂. The inset in (a) is the magnified view of the absorption spectra. **b**, **c**, Transient PL decays of HzTFEX₂ (b) and HzPipX₂ (c) at varying temperatures. **d**, Temperature-dependence of τ_{DF} of HzTFEX₂ and HzPipX₂; the solid lines in (d) represent the fits of τ_{DF} to a single exponential in inverse temperature. **e**, **f**, Schematic diagram of the excited states and the associated transitions of HzTFEX₂ (e) and HzPipX₂ (f).

- In Fig. 3b and 3c, one cannot see the prompt fluorescence. A display on a log-log scale might be more suitable here and is more common in the field of TADF emitters. Also, adding the biexponential fit to it would be useful. Further, I would find it more helpful if you gave the temperature in K, and not in inverted K. There is no reason here to quote temperature in inverted units.

Re: We have included the log-log representation of the transient PL decays as Supplementary Fig. 3:

Supplementary Fig. 3 | Log-log representation of the transient PL decays of HzTFEX₂ and HzPipX₂ at varying temperatures. a, b, Transient PL decays of HzTFEX₂ (a) and HzPipX₂ (b) in deaerated toluene at varying temperatures.

We have also included the detailed results of the biexponential fitting as Supplementary Fig. 6, Supplementary Fig 7, and Supplementary Table 3. We have quoted the temperature in inverted K in Fig. 3b and 3c to relate it with the X axis in Fig 3d.

Overall, I fear that in the present form, the conclusion is not sufficiently supported by the data, and I do not recommend publication at this stage. I feel the topic is very interesting, and the paper reads well, and a carefully revised version might merit consideration.

Re: We thank Referee#2 again for the constructive suggestions. We have carefully revised the manuscript by revealing the data. We hope that Referee#2 will find the revised manuscript and the point-by-point response convincing to validate the conclusion.

Referee #3 (Remarks to the Author):

The article by Aizawa, Pu, Miyajima et al presents some interest in the field of heptazines and TADF molecules, is correctly written, and presents reliable data. However, in my opinion it is far from presenting enough originality and novelty to warrant publication in a top general journal like nature. I precise my point of view hereunder

The discovery of the TADF emission of heptazines was made by C. Adachi in 2013 and 2014, who published almost simultaneously two papers (on symmetrical aryl heptazines). They also demonstrated, in one case with the same external emission yield (EQE) of 17%, the feasibility of a performing OLED.

A little later, W. Domcke published a series of 4 papers detailing the beautiful discovery, through a theoretical approach, of the fact that most heptazines should present themselves with the S1-T1

inversion.

Finally, the synthetic chemistry presented is nothing new, nucleophilic substitution on trichloroheptazine having been extensively documented by E. Kroke (and L. Dubois for amines), while the discovery of the pseudo-electrophilic substitution was discovered much earlier before by E. Kober in 1962. Similarly examples of unsymmetrical heptazines can be found here and there in the literature...

Therefore the only novelty of this article is the idea of screening heptazines using a coarse calculation method, which relies on the (very reasonable) that coarse calculations can be correlated through the entire family, to extract the “interesting” ones. However, the S1-T1 inversion, contrarily to what is stated by the authors, although an amusing exotic feature confirming Domcke’s calculations, is not the key point in this family of compounds...

Re: We respectfully disagree with Referee#3 on this point. The novelty of this work is the experimental demonstration of molecules possessing negative ΔE_{ST} and the resultant delayed fluorescence with anomalous features: (I) the very short decay time constants ($\tau_{DF} \sim 0.2$ microseconds), (II) the decreasing trend of τ_{DF} with lowering the temperature, and (III) the rate inversion of RISC and ISC ($k_{RISC} > k_{ISC}$). In fact, the abstract and the conclusion claim these experimental results.

There are other points requiring concern, the absence of which, also lowers the overall interest of the work. There is first a ground problem with the authors’ approach. It is true that they present the first example of a heptazine with a demonstrated T1-S1 inversion. Meanwhile, none of the (rare) previous authors interested in other TADF heptazines took this into consideration, probably because this is actually not a key point. Adachi did not put interest on the matter, neither P. Audebert, who were the first only groups to raise interest in the field (4 reports altogether). While with the fully aromatic conjugated symmetrical Adachi’s heptazines, I would suspect the triplet to be slightly below the singlet, on the other hand I am convinced that if one would check the temperature-dependence of the fluorescence of Audebert’s alkoxy heptazines (Chem Commun, 2020) one would observe the same S1-T1 inversion, with at least a couple of them. This is because the reported decay times are quite in line with the ones reported here (by the way, the authors of the present report, although for an alleged different reason, also contains a heptazine also with one TFE substituent! A fact that would have certainly deserved to be noticed, and discussed a little...). The reason for these “omissions” in previous reports is likely that, as far as TADF is pursued on, the fact that the triplet lies a little below, at the same level or a little above the singlet is actually unimportant, since extensive TADF will indeed always be observed at 300°K. On the other hand, and as Adachi realized and extensively discussed, the important point is the quantum yield. On this point, the two

heptazines presented are interesting, while again not exceptional, with yields in the 50-70% range, lying in between Adachi's (80-90%) and Audebert's (20-30%) ones. Anyhow, despite the four Adachi's and Audebert's papers are correctly cited, their results must nevertheless be thoroughly discussed in a new version of this work, which has exactly the same points of interest.

Re: We respectfully disagree with Referee#3 on this point. Only from the fluorescence lifetimes, one cannot determine whether the ΔE_{ST} is positive or negative. Please note that the Adachi's heptazines have been reported with positive ΔE_{ST} and sub-millisecond transient EL (Ref. 26, 27). Based on the comments from Referees#1 and #2, there appears to be a consensus on the novelty and significance of the experimental demonstration of negative ΔE_{ST} . As explained in the introduction, conventional TADF materials suffer from the long excited-state lifetimes in the microsecond or millisecond range (even at 300 K). We have mentioned the previous works on page 2, line 62:

“Correlated wave function theories suggested that S_1 of heptazine lies 0.2–0.3 eV below T_1 , though S_1 is a ‘dark’ state, meaning that the electronic transition to the ground state (S_0) is dipole-forbidden and the oscillator strength (f) is zero in the D_{3h} symmetry point group. Interestingly, the heptazine core is shared by several synthesised molecules that exhibit intense TADF^{24,25} with positive ΔE_{ST} ^{26,27}.”

The previous comment takes me to my last important point of concern. Unfortunately, no coarse theoretical approach (at least as far as I know) is able to, even grossly, predict the fluorescence yield of an heptazine, let alone another fluorophore family. But on the other hand, from the authors' data on Fig. 1d, I disagree on the view of a tradeoff between f and ΔE_{ST} . Meanwhile, looking closer at the data on Fig. 1d, it shows indeed a strong correlation (with very few exceptions!) between the main bandgap and the ΔE_{ST} . (said otherwise, most heptazines with (relatively...) split T and S are also blue emitting, while the ones with closer S and T are yellow to red emitting). This is quite interesting indeed, and overlooked in the paper; conversely the rare exceptions should deserve more attention. On the other hand, f seems to be much less influent (on ΔE_{ST}), while it might have huge consequences on the QY, as very frequently observed. This also would make an interesting addition to the article. As blue emitters are highly desirable, and high f values are often connected to high QYs, I would rather look for heptazines in the 0.2-0.3/0.1-0.15 zone of the Fig. 1d graph, where a few blue points are still present...

Re: Fig. 1d shows the main bandgap (S_1 - S_0 energy gap) correlates with not only ΔE_{ST} but also f . Thus, the manuscript have explained this point on page 3, line 89:

“A well-known trade-off between small ΔE_{ST} and large f is evident from this particular data set of heptazine analogues. While balancing such trade-off is a key concern in recent synthetic efforts on TADF materials, Fig. 1c demonstrates the optimal combinations of ΔE_{ST} and f , for which one parameter cannot be improved

anymore without sacrificing the other. Fig. 1d further visualises the trade-off between ΔE_{ST} and f for each fluorescence colour.”

As explained in the manuscript, the introduction the chemical substitutions to heptazine recovered its f and led to HzTFEX₂ and HzPipX₂ with relatively high PL quantum yields (QYs) of 74% and 67%. The heptazines in the zone suggested by Referee#3 are indeed interesting because they have potentials to exhibit further improved QYs. However, the exploration of this point is beyond the scope of this study on the experimental evaluation of the negative ΔE_{ST} and the consequent unique delayed fluorescence.

Minor point: On the four remarkable papers of Domcke on the heptazines' fluorescence, only one is cited, a situation which needs to be corrected when submitting to another journal; A recent 2021 Chem Rev by Kroke, Audebert et al., making the point on molecular heptazines, that the authors probably have missed, must of course also be cited, and its content might in addition help to write down the suggested discussions on the aforementioned points.

Re: We thankfully agree with the relevance of the nice general review on molecular heptazines and have cited it as Ref. 21. We have pursued the following 5 papers by Domcke et al. and have cited I and V as Ref. 15 and 20, respectively. Please note that II, III, and IV are studies on the photocatalysis and/or proton-coupled electron transfer of the compound reported in I.

I. Ehrmaier, J., Rabe, E. J., Pristash, S. R., Corp, K. L., Schlenker, C.W., Sobolewski, A. L. & Domcke, W. Singlet–triplet inversion in heptazine and in polymeric carbon nitrides. *J. Phys. Chem. A* **123**, 8099–8108 (2019).

II. Rabe, E. J., Corp, K. L., Huang, X., Ehrmaier, J., Flores, R. G., Estes, S. L., Sobolewski, A. L., Domcke, W. & Schlenker, C. W. Barrierless Heptazine-Driven Excited State Proton-Coupled Electron Transfer: Implications for Controlling Photochemistry of Carbon Nitrides and Aza-Arenes, *J. Phys. Chem. C* **123**, 29580–29588 (2019).

III. Ehrmaier, J., Huang, X., Rabe, E. J., Corp, K. L., Schlenker, C. W., Sobolewski, A. L. & Domcke, W. Molecular Design of Heptazine-Based Photocatalysts: Effect of Substituents on Photocatalytic Efficiency and Photostability, *J. Phys. Chem. A* **124**, 3698–3710 (2020).

IV. Corp, K. L., Rabe, E. J., Huang, X., Ehrmaier, J., Kaiser, M. E., Sobolewski, A. L., Domcke, W. & Schlenker, C. W. Control of Excited-State Proton-Coupled Electron Transfer by Ultrafast Pump-Push-Probe Spectroscopy in Heptazine-Phenol Complexes: Implications for Photochemical Water Oxidation, *J. Phys. Chem. C* **124**, 9151–9160 (2020).

V. Sobolewski, A. L. & Domcke, W. Are Heptazine-Based Organic Light-Emitting Diode Chromophores Thermally Activated Delayed Fluorescence or Inverted Singlet–Triplet Systems? *J. Phys. Chem. Lett.* **12**, 6852–6860 (2021).

Reviewer Reports on the First Revision:

Referees' comments:

Referee #1 (Remarks to the Author):

The authors have done an important (additional) effort to answer all the (computationally oriented) questions raised. They have now better linked their results with previous theoretical studies, and still more importantly, the set of additional calculations performed has allowed to consistently bracket the values disclosed before.

Therefore, the part dealing with calculations has been qualitatively and quantitatively improved, with additional and valuable information added to the SI. Overall, the answer of the author is largely satisfactory, but I would still like to see if the authors could refine a bit further some of their points:

* I agree about the difficulties for optimizing singlet excited-states by wavefunction methods, but the S1 geometry could be accessed by cost-effective TD-DFT calculations, if needed in future studies.

* The tuning of the w parameter for LC-BLYP employing the 2,5,8-triphenylheptazine should be explained in more detail (e.g., to force matching the HOMO and IP energies) and that information added to the computational methods section.

* The new results by SCS-CC2 and LT-DF-LCC2 could also be mentioned in the manuscript, as well as the dependence of the CASPT2 results with respect to the choice of guess orbitals or the active space size.

* Actually, it is quite satisfying, from my previous experience with related calculations, to see that S1 states are more affected by double-substitutions than T1; which is related to the stabilization of the former vs. the latter, explaining and thus giving possibly rise to the inversion of both states.

Referee #2 (Remarks to the Author):

The manuscript has improved significantly compared to the previous version so that it is easier to follow what the authors did, and I appreciate this. I still think it is addressing an interesting topic, yet I also still think that the discussion and conclusion is not sufficiently supported by the data. It might be possible that the authors' claim of an inverted singlet-triplet gap is correct, yet their evidence is very thin. In fact, I have serious doubts about the correctness and robustness of the conclusion (I shall detail my technical concerns below). Overall, their approach is interesting, yet I feel that a publication in Nature requires, first of all, that the conclusions are reliably supported by robust evidence, and I am afraid this is not the case here. For this reason I regret I cannot recommend the publication of this manuscript in Nature, nor in a related sister journal.

My concern is that the experimental evidence for the key claim of the paper relies fully on the numerical fit of two coupled rate equations to a single PL decay curve. While the authors clarified

the fitting procedure, it is still not transparent to which degree the parameters chosen for the fit can vary, e.g., how big the error bar is on these values compared to the found energy gap of 11 meV. Also, there is an inconsistency in the argument - if S1 is energetically above T1, why should the rate for RISC still be thermally activated? It makes no sense. The observation of a thermal activation energy for RISC in itself contradicts the notion that T1 is above S1. One may then argue that the activation energy reflects a polaronic energy barrier that needs to be overcome for the crossing, akin to a Marcus transfer process. However, if that was the case, then the difference between the activation energy for ISC and RISC could not be used to infer information of the energy gap between the states. In short, I feel the key conclusion of the paper is on very uncertain grounds.

If these difficulties and ambiguities in the analysis of the experimental data are clearly stated, I feel the manuscript might perhaps be of value to a specialized audience that is able to appreciate these intricacies and that can use it as a stimulation for further research and thought in this direction, but I am afraid with all these uncertainties in the correctness of the conclusion it is not suitable for a general audience.

Referee #3 (Remarks to the Author):

The authors have more or less taken into account my suggestions, and now the article is publishable. I noticed another last thing that escaped me initially and would be preferable to correct. This paper is all about heptazines, and the name "heptazine" does not even appear in the title!

I suggest changing the title from:

"Delayed Fluorescence from Inverted Singlet and Triplet Excited States for Efficient Organic Light-Emitting Diodes" into

"Delayed Fluorescence from Inverted Singlet and Triplet Excited States in new heptazines for Efficient Organic Light-Emitting Diodes" or smthg equivalent...

That would be better to find for researchers in the field.

Author Rebuttals to First Revision:

Point-to-Point Answers to Referee#1

The authors have done an important (additional) effort to answer all the (computationally oriented) questions raised. They have now better linked their results with previous theoretical studies, and still more importantly, the set of additional calculations performed has allowed to consistently bracket the values disclosed before.

Therefore, the part dealing with calculations has been qualitatively and quantitatively improved, with additional and valuable information added to the SI. Overall, the answer of the author is largely satisfactory, but I would still like to see if the authors could refine a bit further some of their points:

We highly appreciate these encouraging remarks.

* I agree about the difficulties for optimizing singlet excited-states by wavefunction methods, but the S1 geometry could be accessed by cost-effective TD-DFT calculations, if needed in future studies.

Encouraged by this suggestion from referee#1, we have performed geometry optimization of heptazine by CASSCF and added the results in Supplementary Table 11. We used the 6-31G(d,p) basis set instead of cc-pVDZ since the latter is not compatible with the CASSCF gradient calculations in the MOLPRO program. We did not use TD-DFT because it is not able to account for the double excitation, though it's cost effective.

The following has been included in the revised supporting information.

=====

Adiabatic ΔE_{ST} of heptazine. The S₁ and T₁ geometries of heptazine were optimised by CASSCF(12,12)/6-31G(d,p). The vertical excitation energies and ΔE_{ST} were calculated by CASPT2(12,12)/6-31G(d,p) with a more contracted configuration space⁷. Adiabatic ΔE_{ST} was obtained as the difference between the total CASPT2 energies of the S₁ and T₁ at their optimised geometries. The CASPT2 calculations predicted negative ΔE_{ST} at both S₁ and T₁ geometries, as well as negative adiabatic ΔE_{ST} (Supplementary Table 11).

Supplementary Table 11 | Vertical S_1 and T_1 excitation energies and vertical and adiabatic ΔE_{ST} of heptazine, calculated by CASPT2(12,12)/6-31G(d,p).

Geometry ^a	S_1 excitation energy (eV)	T_1 excitation energy (eV)	Vertical ΔE_{ST} (meV)	Adiabatic ΔE_{ST} (meV)
S_1 geometry	2.305	2.517	-212	-371
T_1 geometry	2.185	2.350	-165	

^aOptimised by the state-averaged (S_0 , S_1 and T_1) CASSCF(12,12)/6-31G(d,p) calculations implemented in the MOLPRO 2019.2 program⁶.

=====

The tuning of the w parameter for LC-BLYP employing the 2,5,8-triphenylheptazine should be explained in more detail (e.g., to force matching the HOMO and IP energies) and that information added to the computational methods section.

We have added the following sentences to explain the details of the computational methods on page 11, line 352 in the revised manuscript.

=====

The range-separation parameter of the LC-BLYP functional was non-empirically optimised to 0.18 bohr^{-1} to minimise the difference between the energy of the HOMO and the ionisation potential of the neutral system and the difference between the energy of HOMO of the radical anion system and the electron affinity of the neutral system⁴⁷ of 2,5,8-triphenylheptazine.

=====

* The new results by SCS-CC2 and LT-DF-LCC2 could also be mentioned in the manuscript, as well as the dependence of the CASPT2 results with respect to the choice of guess orbitals or the active space size.

We have added the following sentence on page 4, line 122 in the revised manuscript.

=====

We note that ΔE_{ST} calculated by other second-order methods are given in Supplementary Information, as well as the dependence of the choice of the guess orbitals and the size of the active space on the CASPT2 results.

=====

Actually, it is quite satisfying, from my previous experience with related calculations, to see that S1 states are more affected by double-substitutions than T1; which is related to the stabilization of the former vs. the latter, explaining and thus giving possibly rise to the inversion of both states.

We greatly appreciate referee#1's constructive suggestions on this work. They have certainly helped us improve our manuscript.

Point-to-Point **Answers** to Referee#2

The manuscript has improved significantly compared to the previous version so that it is easier to follow what the authors did, and I appreciate this. I still think it is addressing an interesting topic, yet I also still think that the discussion and conclusion is not sufficiently supported by the data. It might be possible that the authors' claim of an inverted singlet-triplet gap is correct, yet their evidence is very thin. In fact, I have serious doubts about the correctness and robustness of the conclusion (I shall detail my technical concerns below). Overall, their approach is interesting, yet I feel that a publication in Nature requires, first of all, that the conclusions are reliably supported by robust evidence, and I am afraid this is not the case here. For this reason I regret I cannot recommend the publication of this manuscript in Nature, nor in a related sister journal.

My concern is that the experimental evidence for the key claim of the paper relies fully on the numerical fit of two coupled rate equations to a single PL decay curve.

We respectfully disagree with referee#2 on this point. Our conclusion that S_1 and T_1 of HzTFEX₂ is energetically inverted is not only based on the numerical fit but also the experimental observations. For clarification, we explain how our claim are supported by the experimental observations. After that, we clarify the validity of the numerical fit to further support our claim.

[Experimental Observations That Supports Our Claims]

Fig. 3 b, c, Transient PL decays of HzTFEX₂ (b) and HzPipX₂ (c) at varying temperatures.

As shown in Fig. 3b, the time constant (lifetime) of delayed fluorescence of HzTFEX₂ **decreases** with lowering the temperature. This decreasing trend has never been reported in any other TADF molecules (*i.e.* positive ΔE_{ST} systems). We are confident that this unprecedented observation is originated from the inverted S_1 and T_1 of HzTFEX₂. In fact, the time constant of delayed fluorescence of conventional TADF molecules **increases** with lowering the temperature as in the case of HzPipX₂ (Fig. 3c). This increasing trend is widely recognized in TADF systems and explained as the following: “the populations of S_1 and T_1 reach the steady-state condition by fast ISC and RISC compared to the radiative decay of S_1 . According to thermodynamic laws,

the steady-state population shifts to the energetically lower-lying state with lowering the temperature. For TADF systems with positive ΔE_{ST} , the dark T_1 is energetically lower than the emissive S_1 . As a result, the time constant (lifetime) of conventional TADF increases with lowering temperature.” According to this general understanding of the temperature dependence of TADF systems, the observed opposite trend of HzTFEX₂ strongly supports the inverted S_1 and T_1 . In addition, the transient absorption spectroscopy has revealed that the S_1 and T_1 of HzTFEX₂ decay maintaining the constant population ratio with the equivalent decay time constants in the time range from 100 ns to 350 ns, supporting that the populations of S_1 and T_1 reach the steady-state condition after photoexcitation (Extended Data Fig.1 c).

Extended Data Fig. 1 c Transient absorption decays of S_1 and T_1 monitored at 700 nm and 1600 nm, respectively.

It is also noted that non-radiative decay from S_1 to S_0 is generally suppressed with lowering the temperature and the time constant of conventional fluorescence **increases** with lowering temperatures. Hence, it is clear that the suppression of the non-radiative decay does not affect our claim because the time constant of the delayed fluorescence of HzTFEX₂ **decreases** with lowering temperature. To the best of our knowledge, there is no other mechanism except for the inverted S_1 and T_1 can rationally explain this unprecedented trend.

[Numerical Fit]

The two coupled rate equations used for the numerical fit are the basis of the kinetic analysis of delayed fluorescence. For example, the well-known biexponential PL decay model, $I_{PL} = A_1 \exp(-t/\tau_{PF}) + A_2 \exp(-t/\tau_{DF})$, is deviated from the two coupled rate equations [Dias, F.B. *Phil. Trans. R. Soc. A* **373**, 20140447 (2015), Supporting information in Kaji, H. *et al. Nat. Commun.* **6**, 8476 (2015), Einzinger, M. *et al. Adv. Mater.* **29**, 1701987 (2017)]. Thus, the numerical fit with these equations is appropriate for determining the kinetics of delayed fluorescence and has been widely used [Ref. 50 in the manuscript, Hempe, M. *et al. Chem. Mater.* **33**, 3066 (2021). Stavrou,, K. *et al. ACS Appl. Mater. Interfaces* **13**, 8643 (2021). Gomes, L. *et al. J. Phys. Chem. Lett.* **12**, 1490 (2021), Stavrou,, K. *et al. ACS Appl. Electron. Mater.* **2**, 2868 (2020). Streiter, M. *et al. J. Phys. Chem. C* **124**, 15007 (2020)].

To further address referee#2's concern, we have calculated k_r , k_{nr} , k_{ISC} , and k_{RISC} from the parameters of the biexponential PL decay, A_1 , A_2 , τ_{PF} , and τ_{DF} , by the method reported by Tsuchiya *et al.* [Tsuchiya, Y. *et al. J.*

Phys. Chem. A **125**, 8074 (2021)]. This method also revealed the rate inversion of ISC and RISC (i.e. $k_{\text{RISC}} > k_{\text{ISC}}$), further supporting the energy inversion of S_1 and T_1 . We have added this data as Supplementary Table 12.

=====

Confirming the rate inversion of ISC and RISC with an alternative method. The excited-state kinetics of HzTFEX₂ in deaerated toluene solutions at 300 K was also analysed by the method reported by Tsuchiya *et al.*⁸ As consistent with the results discussed in the main text, Tsuchiya's method also revealed the rate inversion of ISC and RISC (i.e. $k_{\text{RISC}} > k_{\text{ISC}}$).

Supplementary Table 12 | k_{r} , k_{nr} , k_{ISC} and k_{RISC} of HzTFEX₂ determined by different methods.

Method	$k_{\text{r}} (\text{s}^{-1})$	$k_{\text{nr}} (\text{s}^{-1})$	$k_{\text{ISC}} (\text{s}^{-1})$	$k_{\text{RISC}} (\text{s}^{-1})$
Fit using Eq. (1)	5.4×10^6	1.9×10^6	2.3×10^7	4.2×10^7
Tsuchiya's method	5.9×10^6	2.0×10^6	2.6×10^7	4.1×10^7

While the authors clarified the fitting procedure, it is still not transparent to which degree the parameters chosen for the fit can vary, e.g., how big the error bar is on these values compared to the found energy gap of 11 meV.

We thank referee#2 for pointing this out. The standard errors of k_{ISC} and k_{RISC} of HzTFEX₂ are less than $1.8 \times 10^5 \text{ s}^{-1}$ and are two orders of magnitude smaller than the values of k_{ISC} and k_{RISC} . Thus, the significant digits of k_{ISC} and k_{RISC} are valid and the error bars are smaller than the plot size in Extended Data Fig. 3a. We have added this information in the corresponding figure caption.

Stimulated by this question, we have considered the errors of the activation energies determined by the fit with the Arrhenius equation. The standard error of $E_{\text{a,ISC}}$ and $E_{\text{a,RISC}}$ of HzTFEX₂ were calculated as 1.0 meV and 0.8 meV, respectively. Thus, we have modified ΔE_{ST} of HzTFEX₂ to 11 ± 2 meV in the manuscript. We have also added the errors of $E_{\text{a,ISC}}$, $E_{\text{a,RISC}}$, and ΔE_{ST} of the other emitters. Thus, the revised Extended Data Table 1 now appears as follows:

Extended Data Table 1 | Photophysical properties of HzTFEX₂ and HzPipX₂ in deaerated toluene solutions.

Emitter	λ_{PL} (nm) ^a	Φ_{PL} (%) ^b	τ_{PF} (ns) ^c	τ_{DF} (ns) ^d	k_{r} (s ⁻¹) ^e	k_{nr} (s ⁻¹) ^f	k_{ISC} (s ⁻¹) ^g	k_{RISC} (s ⁻¹) ^h	$E_{\text{a,ISC}}$ (meV) ⁱ	$E_{\text{a,RISC}}$ (meV) ^j	ΔE_{ST} (meV) ^k
HzTFEX ₂	449, 476	74	14	217	5.4×10^6	1.9×10^6	2.3×10^7	4.2×10^7	53 ± 1	42 ± 1	-11 ± 2
HzPipX ₂	442, 467	67	7.9	565	6.3×10^6	3.1×10^6	8.9×10^7	2.2×10^7	17	69 ± 1	52 ± 1
HzTFEP ₂	454, 483	44	35	288	3.1×10^6	4.0×10^6	1.4×10^7	1.8×10^7	31 ± 2	17 ± 1	-14 ± 3
HzTFET ₂	451, 479	42	23	246	2.9×10^6	4.0×10^6	1.6×10^7	2.7×10^7	44 ± 2	31 ± 1	-13 ± 3

^aPhotoluminescence (PL) peak wavelength. ^bPL quantum yield. ^cTime constant of prompt fluorescence. ^dTime constant of delayed fluorescence. ^eRate constant of radiative decay of the lowest-energy excited state (S₁) to the ground state (S₀). ^fRate constant of non-radiative decay of S₁ to S₀. ^gRate constant of intersystem crossing (ISC) of S₁ to the lowest-energy triplet excited state (T₁). ^hRate constant of reverse intersystem crossing (RISC) of T₁ to S₁. ⁱActivation energy of ISC. ^jActivation energy of RISC. ^kEnergy gap between S₁ and T₁.

Also, there is an inconsistency in the argument - if S₁ is energetically above T₁, why should the rate for RISC still be thermally activated? It makes no sense. The observation of a thermal activation energy for RISC in itself contradicts the notion that T₁ is above S₁. One may then argue that the activation energy reflects a polaronic energy barrier that needs to be overcome for the crossing, akin to a Marcus transfer process. However, if that was the case, then the difference between the activation energy for ISC and RISC could not be used to infer information of the energy gap between the states. In short, I feel the key conclusion of the paper is on very uncertain grounds.

Fig. R1 Schematic potential energy surfaces of S₁ and T₁ with positive ΔE_{ST} (left) and negative ΔE_{ST} (right).

As shown in Fig R1, the activation energies of ISC ($S_1 \rightarrow T_1$) and RISC ($T_1 \rightarrow S_1$) cannot be negative in both cases of positive and negative ΔE_{ST} values in definition. In fact, positive activation energies of ISC ($S_1 \rightarrow T_1$) have been experimentally observed in conventional TADF systems with positive ΔE_{ST} , in which S_1 is energetically above T_1 [Noda, H. *et al. Nat. Mater.* **18**,1084 (2019). Aizawa, N. *et al. Sci. Adv.* **7**, 5769 (2021). Wada, Y. *et al. Nat. Photonics* **14**, 643 (2020)]. In addition, it is known that the difference between $E_{a,ISC}$ and $E_{a,RISC}$ of a representative TADF molecule 4CzIPN [Uoyama, H. *et al. Nature* **492**, 234 (2012)], for example, is consistent with ΔE_{ST} determined from its fluorescence and phosphorescence spectra [Noda, H. *et al. Nat. Mater.* **18**,1084 (2019)]. It is also noted that phosphorescence of HzTFEX₂ cannot be observed since the phosphorescence competes with fast RISC.

To improve the clarify of this point, we have added Extended Data Fig. 3c showing the ΔE_{ST} is the difference between the activation energies of ISC and RISC. The rational of this is that ISC and RISC are caused by a very weak coupling between the singlet and triplet excited states (*i.e.* spin-orbit coupling of organic molecules) and thus occur at the crossing. The temperature dependence of experimental k_{ISC} and k_{RISC} has been demonstrated to follow the non-adiabatic transition model assuming this crossing [Noda, H. *et al. Nat. Mater.* **18**,1084 (2019). Aizawa, N. *et al. Sci. Adv.* **7**, 5769 (2021). Wada, Y. *et al. Nat. Photonics* **14**, 643 (2020)]. In this case, the activation energy is the energy required to reach the molecular geometry where the singlet and triplet excited states possess equivalent energies, rather than the polaronic energy barrier modeled in the Marcus theory for non-adiabatic electron transfer reactions.

=====

Extended Data Fig. 3 | k_{ISC} and k_{RISC} of HzTFEX₂ and HzPipX₂. **a, b**, Temperature-dependence of k_{ISC} and k_{RISC} of HzTFEX₂ (a) and HzPipX₂ (b) in deaerated toluene. The solid lines in (a) and (b) represent the fits of the plots to the Arrhenius equation. **c**, Schematic diagram of the potential energy surfaces of S_1 and T_1 and the activation energies of ISC and RISC.

=====

If these difficulties and ambiguities in the analysis of the experimental data are clearly stated, I feel the manuscript might perhaps be of value to a specialized audience that is able to appreciate these intricacies and that can use it as a stimulation for further research and thought in this direction, but I am afraid with all these uncertainties in the correctness of the conclusion it is not suitable for a general audience.

We would like to thank referee#2 for taking the time and effort necessary to review our manuscript. Through answering referee#2's questions, our claim of the inverted S_1 and T_1 has been further supported by the robust evidence. We would like to point out again that the observed anomalous features of the delayed fluorescence of HzTFEX₂, (I) the very short decay time constants ($\tau_{DF} \sim 0.2 \mu\text{s}$), (II) the decreasing trend of τ_{DF} with lowering the temperature, (III) $k_{RISC} > k_{ISC}$, (IV) $E_{a,ISC} > E_{a,RISC}$, and (V) the population of $S_1 >$ the population of T_1 , are fully consistent with our conclusion of the inverted S_1 and T_1 .

We believe that the discovery of the molecules that disobey the famous Hund's rule and possess negative ΔE_{ST} will be of great interest to a general audience. In the first round of the review, referee#1 and referee#2 have acknowledged the novelty and significance of our work as following:

“the topic is very exciting and would warrant publication in Nature”

“It is a very important topic for two reasons. From a fundamental science point of view, it is experimental evidence that Hund's rule can be broken in certain circumstances. From an applied science view, it allows for the fabrication of OLEDs with 100% internal quantum efficiency without use of heavy metals or thermally activated transfer from a triplet state.”

“Overall, the experimental confirmation of a negative ΔE_{ST} energy difference is undoubtedly a major breakthrough for the field, with the multi-step computational protocol the first step needed to clearly discard or identify a small but reasonable set of promising candidates. This is really a fascinating field, defying the conventional guidelines, and the contribution of the present manuscript could find definitively its place.”

Point-to-Point Answers to Referee#3

The authors have more or less taken into account py suggestions, and now the article is publishable. I noticed another last thing that escaped me initially and would be preferable to correct. This paper is all about heptazines, and the name "heptazine" does not even appear in the title!

I suggest changing the title from:

"Delayed Fluorescence from Inverted Singlet and Triplet Excited States for Efficient Organic Light-Emitting Diodes" into

"Delayed Fluorescence from Inverted Singlet and Triplet Excited States in new heptazines for Efficient Organic Light-Emitting Diodes" or smthg equivalent...

That would be better to find for researchers in the field.

We greatly appreciate the suggestion form referee#3. Although we well understand the intent of this suggestion to modify the title of the manuscript, the present title also sounds appropriate for us to reach a broad audience. We would like to hear the thoughts of the editors and revise it if needed.

Reviewer Reports on the Second Revision:

Referees' comments:

Referee #4 (Remarks to the Author):

I read the previous referee's report and the author's response letter. In my impression, although the photophysical data are impressive, the scientific validity is not still reliable, and the general impact of the present manuscript has not yet reached the criteria for publication in Nature. In particular, experimental data to support the claim that breaking of Hund's multiplicity rule must be gathered more carefully.

Remark to the Authors:

- 1) O₂ quenching: The authors seem to be convinced that the atmospheric oxygen quenches triplet excitons. This is true in some respects, but it should be noted that the singlet exciton is also quenched by oxygen (please refer: Pure Appl. Chem. 1964, 9, 507, J. Am. Chem. Soc. 1993, 115, 5180, J. Phys. Chem. Lett. 2020, 11, 2, 562). Actually, in supplementary Fig.8, both emission decay components have been attenuated. Therefore, these arguments (i.e., deaerated vs aerated) do not provide direct evidence that the long-lived component originates from the excited spin-triplet. I do think that a transient electron spin resonance measurement can deliver reliable evidence to reinforce the claim.
- 2) TA decay curves at low-temperature: The transient absorption decay curves at low-temperature are somewhat odd. The TA absorbance is related to a density of state at the excited-state. At 200K, the TA absorbance at 1,600 nm disappeared after 250 nano-seconds. However, although the authors claimed that the delayed component originates from the excited spin-triplet, the TA absorbance at 700 nm continues to be observed. It is puzzling.
- 3) Transient decay profiles (Fig.3 b): It is also puzzling why the second decay component is not so sensitive to temperature. I agree that an activation barrier from an initial state to a different state is existence in HzTFEX2, and convince that the second decay component completely disappears at low-temperature region (~10 K). The authors must provide the data. Further, I'd like to recommend that the authors should provide a temperature dependence of radiative decay rate constant from excited spin-singlet (kr).
- 4) PL characteristics in solid: There is no data and discussion regarding the PL characteristics in solid-state. That is not welcome.
- 5) Transient EL decays: The authors mentioned that "HzTFEX2 exhibited the fast transient EL decay reflecting the sub-microsecond H-type delayed fluorescence (Fig.4c)". I do think the authors need to make the argument more carefully. In Fig.4c, the relatively long (microsecond order) delayed component has been observed even in the HzTFEX2 based OLED. Since a negative voltage (-4 V) is applied to eliminate the effect of the trapped charge, it can be judged that this long-life component is not derived from charge recombination. I realized also that the EQE has significantly dropped (~5%) at this on-voltage condition (8 V). So, I should request more transient EL decay data obtained at various voltage (luminance) conditions to dispel my concern. I believe that the authors have realized that similar behavior has been reported in previous literature (Appl. Phys. Lett. 105, 013301 (2014)).
- 6) Benefits of HzTFEX2 for OLED applications: There are a lot of works regarding highly efficient blue

OLEDs including not only TADF but also phosphorescence. However, the device operational lifetime of all of them seems to be fireworks. A most critical point in the research fields is the dramatic improvement of OLED stability. The authors emphasized this point too. However, although it is a critical point, they do not unveil a device operational lifetime of their cases. The suppression of EL efficiency roll-off is also required to use the OLED as practical applications. Since the stability and EQE roll-off strongly depend on the triplet exciton density, we can be expected that the short exciton lifetime of the emitter can strongly contribute to improving these critical characteristics. However, the authors have not been able to present their expected performance. So, what is the advantage of the emitters for OLED application? The authors mentioned that a future optimization of molecular design will unveil the performances. But readers are waiting for the results. Thus, the impact of this paper on the general readers and the researchers in related fields falls short of the criteria required for publication in this journal from the viewpoint of application.

Technical comments to the Authors:

1) Supplementary Fig.5d: x-axis Luminance -> Current density

Referee #5 (Remarks to the Author):

The paper makes an astonishing claim of a material in which the triplet state is at higher energy than the singlet. This is very unusual as it challenges normal teaching in photochemistry and photophysics, and is also potentially useful for organic light-emitting diodes. Hence, if correct, there is no doubt it is suitable for Nature.

I have therefore looked carefully at its central claim and the data supporting it. This centres on two materials, one of which shows normal TADF and the other interpreted as inverted singlet and triplet energy levels (fig 3b of the manuscript). The differences between the decay curves in this figure are small i.e. to first order the decays are rather similar. However, there is a small temperature dependence that is just visible in figure 3d. Surprisingly the lifetime gets shorter and the temperature is decreased – the opposite of a normal TADF material. The authors proceed to interpret this result as TADF with $\Delta E_{ST} < 0$. Overall this does give a consistent interpretation of the data.

The authors should consider if there are any other possible interpretations. For example, the small changes in lifetime could potentially arise from a change in radiative rate as the temperature changes (as noted below the lifetime is not so far from the radiative lifetime). A comment should be added to explain why this is less favorable as an explanation. Overall it does seem difficult to think of alternative explanations that could also account for the reported external quantum efficiency of the OLED.

The authors should do more to explain that their work is a rather special case because the material they have studied has an exceptionally low radiative rate (at least 20 times lower than a typical dye) and it is this very low radiative rate that enables intersystem crossing to compete with radiative decay, and hence enables there to be a significant triplet population that affects the decay kinetics. The delay in fluorescence is actually rather small – the reciprocal of the radiative lifetime is

approximately 200 ns which is the observed lifetime. [I appreciate that taking account of non-radiative decay would reduce this a little, but the situation is still very different from usual TADF materials].

There is an experimental aspect of the study and indeed the logic of the paper that is incomplete. The key photophysical study is done in solution, but the critical evidence of triplet harvesting (in an OLED) is done in the solid phase in a host matrix. It is essential that further photophysical studies are done (like figure 3 b and d) to show that the singlet and triplet are inverted in the solid matrix used in the device as well as in solution.

Subject to all the above points being addressed, I believe this would be a good contribution to Nature.

Author Rebuttals to Second Revision:

Point-to-Point Answers to Referee#4

I read the previous referee's report and the author's response letter. In my impression, although the photophysical data are impressive, the scientific validity is not still reliable, and the general impact of the present manuscript has not yet reached the criteria for publication in Nature. In particular, experimental data to support the claim that breaking of Hund's multiplicity rule must be gathered more carefully.

We would like to thank the referee#4 for taking the time and effort necessary to review our manuscript. The below valuable and constructive suggestions have certainly helped us improve our manuscript.

Remark to the Authors:

1) O₂ quenching: The authors seem to be convinced that the atmospheric oxygen quenches triplet excitons. This is true in some respects, but it should be noted that the singlet exciton is also quenched by oxygen (please refer: Pure Appl. Chem. 1964, 9, 507, J. Am. Chem. Soc. 1993, 115, 5180, J. Phys. Chem. Lett. 2020, 11, 2, 562). Actually, in supplementary Fig.8, both emission decay components have been attenuated. Therefore, these arguments (i.e., deaerated vs aerated) do not provide direct evidence that the long-lived component originates from the excited spin-triplet. I do think that a transient electron spin resonance measurement can deliver reliable evidence to reinforce the claim.

Our claim that the long-lived emission is delayed fluorescence originated from reversible ISC and RISC between S₁ and T₁ is not only supported by the observed quenching by O₂ but also the transient absorption decays (Extended Data Fig. 1c). The below explanation can be found on page 4, line 142 in the manuscript. In addition, the maximum external quantum efficiency of the fabricated OLED reached 17% proving the excited spin-triplet state contributing to the light emission.

Extended Data Fig. 1 c, Transient absorption decays of S₁ and T₁ monitored at 700 nm and 1600 nm, respectively.

(Page 4, line 142 in the manuscript)

Since atmospheric O_2 can quench molecular triplet excited states and the change in Φ_{PL} is reversible, we ascribe the blue emissions of the two molecules, at least partially, to delayed fluorescence through forward intersystem crossing (ISC) and RISC between S_1 and T_1 . This assumption is supported by transient absorption decay measurements on HzTFEX₂, which probed ISC from S_1 to T_1 as the signal decay of S_1 at 700 nm and the signal growth of T_1 at 1600 nm, followed by the persistent signal decays of both S_1 and T_1 (Extended Data Fig. 1). We also note that both decays have similar time constants (223 ns for S_1 and 210 ns for T_1), indicating the steady-state condition with the constant population ratio maintained by ISC and RISC.

Supplementary Fig 8 shows the transient absorption decays of S_1 and T_1 monitored at 700 nm and 1600 nm, respectively. We explained that the quenching of the T_1 by O_2 resulted in the attenuated S_1 decay due to the interconversion of S_1 and T_1 by ISC and RISC. However, as the referee#4 pointed out, we cannot deny the possibility that the S_1 was also directly quenched by O_2 . Thus, we have included the following sentences and the suggested literatures as Ref. 4–6 in the supplementary information.

Additional transient absorption measurements. We performed transient absorption measurements on HzTFEX₂ in deaerated and aerated toluene solutions (Supplementary Fig. 9). The transient absorption of T_1 monitored at 1600 nm decayed faster in the aerated solution than the deaerated solution, indicating the quenching the T_1 by atmospheric O_2 . The quenching of the T_1 accelerated the S_1 decay monitored at 700 nm due to the interconversion of S_1 and T_1 by ISC and RISC. We also note the S_1 can be also quenched directly by O_2 ^{4–6}.

Supplementary Fig. 9 | Transient absorption of HzTFEX₂ in deaerated and aerated toluene solutions. Transient absorption decays of S_1 and T_1 monitored at 700 nm and 1600 nm, respectively, in deaerated and aerated toluene solutions.

We agree with the referee#4 that transient electron spin resonance (TrESR) measurements can further reinforce our claim by probing the triplet excitons. We have started discussion with experts in TrESR and found that such investigations are best to be conducted as a different study because well-resolved TrESR spectra are generally available only at very low temperatures. We would like to conduct TrESR studies to reveal the detailed formation dynamics of the triplet excitons in the emitters and present them in different papers.

2) TA decay curves at low-temperature: The transient absorption decay curves at low-temperature are somewhat odd. The TA absorbance is related to a density of state at the excited-state. At 200K, the TA absorbance at 1,600 nm disappeared after 250 nano-seconds. However, although the authors claimed that the delayed component originates from the excited spin-triplet, the TA absorbance at 700 nm continues to be observed. It is puzzling.

The disappearance of the TA signal at 1600 nm does not necessarily mean that the T_1 density is completely zero because there are the limits of detection for the lowest signal at each wavelength. We also note that the absorption coefficient of T_1 does not necessarily equal to that of S_1 .

3) Transient decay profiles (Fig.3 b): It is also puzzling why the second decay component is not so sensitive to temperature. I agree that an activation barrier from an initial state to a different state is existence in HzTFEX₂, and convince that the second decay component completely disappears at low-temperature region (~10 K). The authors must provide the data. Further, I'd like to recommend that the authors should provide a temperature dependence of radiative decay rate constant from excited spin-singlet (kr).

The relatively small sensitivity of the second decay component (delayed fluorescence) to temperature corresponds to the small ΔE_{ST} of HzTFEX₂ (-11 meV). We have performed additional transient PL measurements on HzTFEX₂ at a very low temperature of 80 K. We used PPF, bis(diphenylphosphoryl)dibenzo[*b,d*]furan, as a host matrix to reach that low temperature (below the melting point of toluene). As the referee#4 expected, the second decay component almost disappeared because the activation energy is too high to generate T_1 at 80 K. We have included the following sentences and Supplementary Fig. 12 in the revised supplementary information.

The delayed fluorescence of PPF:1wt% HzTFEX₂ was suppressed at 80 K (Supplementary Fig. 12). This behaviour suggests that ISC is much slower than the radiative decay of S₁ and the T₁ population is small at very low temperatures where the thermal energy is too low to overcome $E_{a,ISC}$.

Supplementary Fig. 12 | Transient PL decays of HzTFEX₂ in a PPF host matrix at 300 K and 80 K. Transient PL decays of PPF:1wt% HzTFEX₂ at 300 K and 80 K under a N₂ atmosphere.

We have also added the temperature dependence of $k_r + k_{nr}$ as Supplementary Fig. 4 to the revised supplementary information. The change in $k_r + k_{nr}$ is negligible compared to the changes in k_{ISC} and k_{RISC} (Extended Data Fig. 3). Thus the decreasing trend of τ_{DF} of HzTFEX₂ is more reasonably attributed to the inverted S₁ and T₁.

Supplementary Fig. 4 | Rate constants of radiative decay and non-radiative decay (k_r and k_{nr}). Temperature dependence of $k_r + k_{nr}$ of HzTFEX₂ and HzPipX₂ determined by numerically fitting Eq. (1) in the main text to the transient PL decays measured in deaerated toluene solutions.

Extended Data Fig. 3 | k_{ISC} and k_{RISC} of HzTFEX₂ and HzPipX₂. **a, b,** Temperature-dependence of k_{ISC} and k_{RISC} of HzTFEX₂ (a) and HzPipX₂ (b) in deaerated toluene. The solid lines in (a) and (b) represent the fits of the plots to the Arrhenius equation. The error bars of the plots in (a) and (b) are smaller than the plot size. **c,** Schematic diagram of the potential energy surfaces of S₁ and T₁ and the activation energies of ISC and RISC.

4) PL characteristics in solid: There is no data and discussion regarding the PL characteristics in solid-state. That is not welcome.

We have added the following PL characteristics of HzTFEX₂ in a solid state in the revised supplementary information.

Photophysical properties of HzTFEX₂ in a solid host matrix. The photophysical properties of HzTFEX₂ were evaluated in a PPF host matrix under a N₂ atmosphere. Upon photoexcitation a thin film of PPF:1wt% HzTFEX₂ emits blue emission with two peaks (λ_{PL}) at 442 nm and 470 nm, respectively (Supplementary Fig. 11a) and a PL quantum yield (Φ_{PL}) of 86%. The transient PL decays comprise nanosecond-order prompt fluorescence followed by sub-microsecond delayed fluorescence (Supplementary Fig. 11b). The time constant of delayed fluorescence (τ_{DF}) gradually decreases from 207 ns to 146 ns with lowering the temperature from 300 K to 143 K (Supplementary Fig. 11c). By numerically fitting Eq. (1) in the main text to the transient PL decays at 300 K, k_{ISC} and k_{RISC} were determined to be $2.6 \times 10^7 \text{ s}^{-1}$ and $2.8 \times 10^7 \text{ s}^{-1}$. $E_{a,ISC}$ and $E_{a,RISC}$ were extracted from the temperature dependence of k_{ISC} and k_{RISC} to be $21 \pm 1 \text{ meV}$ and 12 meV (Supplementary Fig. 11d). ΔE_{ST} was determined to be $-8 \pm 1 \text{ meV}$ by subtracting $E_{a,ISC}$ from $E_{a,RISC}$.

Supplementary Fig. 11 | Photophysical properties of HzTFEX₂ in a PPF host matrix. **a**, Steady-state PL spectra of a thin film of PPF:1wt% HzTFEX₂. **b**, Transient PL decays of PPF:1wt% HzTFEX₂ at varying temperatures under a N₂ atmosphere. **c**, Temperature-dependence of τ_{DF} of PPF:1wt% HzTFEX₂. **d**, Temperature-dependence of k_{ISC} and k_{RISC} of PPF:1wt% HzTFEX₂. The solid lines in (d) represent the fits of the plots to the Arrhenius equation.

5) Transient EL decays: The authors mentioned that “HzTFEX₂ exhibited the fast transient EL decay reflecting the sub-microsecond H-type delayed fluorescence (Fig.4c)”. I do think the authors need to make the argument more carefully. In Fig.4c, the relatively long (microsecond order) delayed component has been observed even in the HzTFEX₂ based OLED. Since a negative voltage (−4 V) is applied to eliminate the effect of the trapped charge, it can be judged that this long-life component is not derived from charge recombination. I realized also that the EQE has significantly dropped (~5%) at this on-voltage condition (8 V). So, I should request more transient EL decay data obtained at various voltage (luminance) conditions to dispel my concern. I believe that the authors have realized that similar behavior has been reported in previous literature (Appl. Phys. Lett. 105, 013301 (2014)).

We have included the additional transient EL decay data in Supplementary Fig. 13. The comparison of the transient EL decays at different on-voltages suggest that the long-life component could be caused by bimolecular recombination rather than recombination of the trapped charges. The literature suggested by the referee#4 also reported transient EL decays with a long-life component, though it was in the millisecond time range.

Transient electroluminescence (EL) decay at different voltages. Transient EL decays of the OLED using HzTFEX₂ were measured in pulse operation with square-wave voltages steps from 5 V to -4 V and from 8 V to -4V, respectively. In both voltage conditions, the device exhibited sub-microsecond transient EL decays, as well as relatively weak and long decays in the microsecond time range. Since the negative voltage of -4 V was applied, the recombination of the trapped charges should play a minor role in the long decays. In addition, the long decay component was enhanced at the higher on-voltage (i.e., higher exciton density). It is thus not unreasonable to conclude that the long decay component was caused by bimolecular recombination.

Supplementary Fig. 13 | Transient EL decays of the OLED using HzTFEX₂. Transient EL decays of the OLED using HzTFEX₂ measured in pulse operation with square-wave voltages steps from 5 V to -4 V and from 8 V to -4V, respectively.

6) Benefits of HzTFEX₂ for OLED applications: There are a lot of works regarding highly efficient blue OLEDs including not only TADF but also phosphorescence. However, the device operational lifetime of all of them is seems to be fireworks. A most critical point in the research fields is the dramatic improvement of OLED stability. The authors emphasized this point too. However, although it is a critical point, they do not unveil a device operational lifetime of their cases. The suppression of EL efficiency roll-off is also required to use the OLED as practical applications. Since the stability

and EQE roll-off strongly depend on the triplet exciton density, we can be expected that the short exciton lifetime of the emitter can strongly contribute to improving these critical characteristics. However, the authors have not been able to present their expected performance. So, what is the advantage of the emitters for OLED application? The authors mentioned that a future optimization of molecular design will unveil the performances. But readers are waiting for the results. Thus, the impact of this paper on the general readers and the researchers in related fields falls short of the criteria required for publication in this journal from the viewpoint of application.

We agree with the referee#4 that the device operational lifetime is an critical point for practical implications. However, stability of OLEDs is a very complex problem, and one which is difficult to delve into much detail in this manuscript. Peripheral materials and device structures used have a significant impact on the degradation mechanism and thus the operational lifetime. In fact, a representative green TADF material, 4CzIPN, was first reported in 2012 [Uoyama et al. *Nature* **492**, 234–238 (2012)] and afterwards its device operational lifetime was reported with completely different peripheral materials and device structures [Nakanotani et al. *Sci. Rep.* **3**, 2127 (2013)].

Technical comments to the Authors:

1) Supplementary Fig.5d: x-axis Luminance -> Current density

We have thankfully corrected the x-axis in Supplementary Fig. 6d. The corrected Supplementary Fig. 6 now appears as below

Supplementary Fig. 6 | OLED performance. a, b, c, d, Electroluminescence (EL) spectra measured at 1.0 mA (a), current density–voltage characteristics (b), luminance–voltage characteristics (c), and external quantum efficiency–current density characteristics (d) of the Device I, II, and III.

Point-to-Point Answers to Referee#5

Referee #5 (Remarks to the Author):

The paper makes an astonishing claim of a material in which the triplet state is at higher energy than the singlet. This is very unusual as it challenges normal teaching in photochemistry and photophysics, and is also potentially useful for organic light-emitting diodes. Hence, if correct, there is no doubt it is suitable for Nature.

I have therefore looked carefully at its central claim and the data supporting it. This centres on two materials, one of which shows normal TADF and the other interpreted as inverted singlet and triplet energy levels (fig 3b of the manuscript). The differences between the decay curves in this figure are small i.e. to first order the decays are rather similar. However, there is a small temperature dependence that is just visible in figure 3d. Surprisingly the lifetime gets shorter and the temperature is decreased – the opposite of a normal TADF material. The authors proceed to interpret this result as TADF with $\Delta E_{ST} < 0$. Overall this does give a consistent interpretation of the data.

We thank the referee#5 for his/her encouraging remarks. We also appreciate the valuable suggestions for improving our manuscript.

The authors should consider if there are any other possible interpretations. For example, the small changes in lifetime could potentially arise from a change in radiative rate as the temperature changes (as noted below the lifetime is not so far from the radiative lifetime). A comment should be added to explain why this is less favorable as an explanation. Overall it does seem difficult to think of alternative explanations that could also account for the reported external quantum efficiency of the OLED.

We have also added the temperature dependence of $k_r + k_{nr}$ as Supplementary Fig. 4 to the revised supplementary information. The change in $k_r + k_{nr}$ is negligible compared to the changes in k_{ISC} and k_{RISC} . Thus the decreasing trend of τ_{DF} of HzTFEX₂ is more reasonably attributed to the inverted S₁ and T₁.

Supplementary Fig. 4 | Rate constants of radiative decay and non-radiative decay (k_r and k_{nr}). Temperature dependence of $k_r + k_{nr}$ of HzTFEX₂ and HzPipX₂ determined by numerically fitting Eq. (1) in the main text to the transient PL decays measured in deaerated toluene solutions.

To explain this point, we have included the following sentence on page 7, line 191 in the revised manuscript.

We note that the change in $k_r + k_{nr}$ at varying temperatures is negligible compared to those in k_{ISC} and k_{RISC} (Supplementary Fig. 4) and thus the decreasing trend of τ_{DF} of HzTFEX₂ is more reasonably attributed to the inverted S₁ and T₁.

The authors should do more to explain that their work is a rather special case because the material they have studied has an exceptionally low radiative rate (at least 20 times lower than a typical dye) and it is this very low radiative rate that enables intersystem crossing to compete with radiative decay, and hence enables there to be a significant triplet population that affects the decay kinetics. The delay in fluorescence is actually rather small – the reciprocal of the radiative lifetime is approximately 200 ns which is the observed lifetime. [I appreciate that taking account of non-radiative decay would reduce this a little, but the situation is still very different from usual TADF materials].

To elucidate this point, we have included the following sentence on page 7, line 210 in the revised manuscript.

In common with the three materials, ISC from S_1 to T_1 competes with the inherently slow radiative decay of heptazines, followed by faster RISC leading to a significant S_1 population relative to T_1 and sub-microsecond DFIST.

We agree with the referee#5 that the ISC from S_1 to T_1 proceeds faster than the slow radiative decay of heptazines, resulting in a significant T_1 population that affects the decay kinetics. We note that k_r of conventional TADF materials is also slow and typically 10^6 – 10^7 s^{-1} .

The reciprocal of k_r (5.4×10^6 s^{-1}) + k_{nr} (1.9×10^6 s^{-1}) is 137 ns and the lifetime of the delayed fluorescence of 217 ns at 300 K. As pointed out by the referee#5, this situation is very different from conventional TADF materials and is indeed due to the inverted S_1 and T_1 enabling $k_{RISC} > k_{ISC}$ (Extended Data Table 1).

Extended Data Table 1 | Photophysical properties of HzTFEX₂ and HzPipX₂ in deaerated toluene solutions.

Emitter	λ_{PL} (nm) ^a	Φ_{PL} (%) ^b	τ_{PF} (ns) ^c	τ_{DF} (ns) ^d	k_r (s ⁻¹) ^e	k_{nr} (s ⁻¹) ^f	k_{ISC} (s ⁻¹) ^g	k_{RISC} (s ⁻¹) ^h	$E_{a,ISC}$ (meV) ⁱ	$E_{a,RISC}$ (meV) ^j	ΔE_{ST} (meV) ^k
HzTFEX ₂	449, 476	74	14	217	5.4×10^6	1.9×10^6	2.3×10^7	4.2×10^7	53 ± 1	42 ± 1	-11 ± 2
HzPipX ₂	442, 467	67	7.9	565	6.3×10^6	3.1×10^6	8.9×10^7	2.2×10^7	17	69 ± 1	52 ± 1
HzTFEP ₂	454, 483	44	35	288	3.1×10^6	4.0×10^6	1.4×10^7	1.8×10^7	31 ± 2	17 ± 1	-14 ± 3
HzTFET ₂	451, 479	42	23	246	2.9×10^6	4.0×10^6	1.6×10^7	2.7×10^7	44 ± 2	31 ± 1	-13 ± 3

^aPhotoluminescence (PL) peak wavelength. ^bPL quantum yield. ^cTime constant of prompt fluorescence. ^dTime constant of delayed fluorescence. ^eRate constant of radiative decay of the lowest-energy excited state (S_1) to the ground state (S_0). ^fRate constant of non-radiative decay of S_1 to S_0 . ^gRate constant of intersystem crossing (ISC) of S_1 to the lowest-energy triplet excited state (T_1). ^hRate constant of reverse intersystem crossing (RISC) of T_1 to S_1 . ⁱActivation energy of ISC. ^jActivation energy of RISC. ^kEnergy gap between S_1 and T_1 .

There is an experimental aspect of the study and indeed the logic of the paper that is incomplete. The key photophysical study is done in solution, but the critical evidence of triplet harvesting (in an OLED) is done in the solid phase in a host matrix. It is essential that further photophysical studies are done (like figure 3 b and d) to show that the singlet and triplet are inverted in the solid matrix used in the device as well as in solution.

We have added the following photophysical properties of HzTFEX₂ in a solid host matrix in the revised supplementary information. The new data validate the inverted S₁ and T₁ of HzTFEX₂ in the solid matrix.

=====
Photophysical properties of HzTFEX₂ in a solid host matrix. The photophysical properties of HzTFEX₂ were evaluated in a PPF host matrix under a N₂ atmosphere. Upon photoexcitation a thin film of PPF:1wt% HzTFEX₂ emits blue emission with two peaks (λ_{PL}) at 442 nm and 470 nm, respectively (Supplementary Fig. 11a) and a PL quantum yield (Φ_{PL}) of 86%. The transient PL decays comprise nanosecond-order prompt fluorescence followed by sub-microsecond delayed fluorescence (Supplementary Fig. 11b). The time constant of delayed fluorescence (τ_{DF}) gradually decreases from 207 ns to 146 ns with lowering the temperature from 300 K to 143 K (Supplementary Fig. 11c). By numerically fitting Eq. (1) in the main text to the transient PL decays at 300 K, k_{ISC} and k_{RISC} were determined to be $2.6 \times 10^7 \text{ s}^{-1}$ and $2.8 \times 10^7 \text{ s}^{-1}$. $E_{\text{a,ISC}}$ and $E_{\text{a,RISC}}$ were extracted from the temperature dependence of k_{ISC} and k_{RISC} to be $21 \pm 1 \text{ meV}$ and 12 meV (Supplementary Fig. 11d). ΔE_{ST} was determined to be $-8 \pm 1 \text{ meV}$ by subtracting $E_{\text{a,ISC}}$ from $E_{\text{a,RISC}}$.

Supplementary Fig. 11 | Photophysical properties of HzTFEX₂ in a PPF host matrix. **a**, Steady-state PL spectra of a thin film of PPF:1wt% HzTFEX₂. **b**, Transient PL decays of PPF:1wt% HzTFEX₂ at varying temperatures under a N₂ atmosphere. **c**, Temperature-dependence of τ_{DF} of PPF:1wt% HzTFEX₂. **d**, Temperature-dependence of k_{ISC} and k_{RISC} of PPF:1wt% HzTFEX₂. The solid lines in (d) represent the fits of the plots to the Arrhenius equation.

Subject to all the above points being addressed, I believe this would be a good contribution to Nature.

We thank the referees again for taking the time and effort necessary to review our manuscript. Their valuable comments/suggestions have certainly helped us improve our manuscript.

Reviewer Reports on the Third Revision:

Referees' comments:

Referee #4 (Remarks to the Author):

The manuscript has been substantially revised and, all of my comments have been addressed and answered convincingly. Thus, I can now recommend publication of the revised manuscript.

Referee #5 (Remarks to the Author):

The authors have made a serious attempt to respond to the comments of the referees, and have provided additional data requested for photophysics of films. This was needed to make the paper logical by linking the photophysics to the device results.

There are some features of the new data I don't understand and these are relevant to assessing its validity. One point is that new supplementary fig 11d is qualitatively different from fig 3a in that the rates and RISC and ISC converge at room temperature (i.e. the red and blue lines intersect in fig S11d, but not in fig 3a. Is this because of lower ΔE_{ST} ?

The second and more important point is that the reported ΔE_{ST} for the film data is -8 meV, whereas in fig 3a it is -11 meV. However, the temperature dependence of the decays in new fig S11 b and c is stronger than for the solution data in fig 3. This seems inconsistent with the smaller magnitude of the activation energy and raises concerns that the methods used do not determine activation energies accurately. I invite the authors to explain this to me.

If the answer to the above points is satisfactory, it is essential to refer to the film results in the main paper between the solution and device results. This could be comments on data in SI, but it's important to state ΔE_{ST} for the film and link it to the supplementary data.

Author Rebuttals to Third Revision:

Point-to-Point Answers to Referee#4

The manuscript has been substantially revised and, all of my comments have been addressed and answered convincingly. Thus, I can now recommend publication of the revised manuscript.

We would like to thank the referee#4 for taking the time and effort necessary to review our manuscript. The below valuable and constructive suggestions have certainly helped us improve our manuscript.

Point-to-Point Answers to Referee#5

Referee #5 (Remarks to the Author):

The authors have made a serious attempt to respond to the comments of the referees, and have provided additional data requested for photophysics of films. This was needed to make the paper logical by linking the photophysics to the device results.

There are some features of the new data I don't understand and these are relevant to assessing its validity. One point is that new supplementary fig 11d is qualitatively different from fig 3a in that the rates of RISC and ISC converge at room temperature (i.e. the red and blue lines intersect in fig S11d, but not in fig 3a. Is this because of lower ΔE_{ST} ?

Yes. Since ΔE_{ST} is the main origin of the difference of k_{ISC} and k_{RISC} , lower ΔE_{ST} leads to similar k_{ISC} and k_{RISC} especially at higher temperatures. We note that k_{RISC} ($2.8 \times 10^7 \text{ s}^{-1}$) is still slightly higher than k_{ISC} ($2.6 \times 10^7 \text{ s}^{-1}$) in the film at room temperature, being consistent with the $k_{RISC} > k_{ISC}$ trend observed in the solution.

The second and more important point is that the reported ΔE_{ST} for the film data is -8 meV, whereas in fig 3a it is -11 meV. However, the temperature dependence of the decays in new fig S11 b and c is stronger than for the solution data in fig 3. This seems inconsistent with the smaller magnitude of the

activation energy and raises concerns that the methods use do not determine activation energies accurately. I invite the authors to explain this to me.

As the pointed by the referee#5, a simple Arrhenius-type equation $k_{DF} = A \exp(-\Delta E_{ST}/k_B T)$, where k_{DF} is the reciprocal of the time constant of delayed fluorescence, gives ΔE_{ST} of -4 ± 1 meV and -7 ± 1 meV for the solution and the film, respectively. However, this approach neglects the temperature dependence of k_r and k_{nr} . Thus, ΔE_{ST} obtained as the difference between the activation energies of ISC and RISC should be more accurate.

We summarise the data related to this point in Fig. R1. Since we have tested the film sample at lower temperatures below the melting point of toluene, at a glance the temperature dependence of the transient PL decays of the film (Fig. R1b) seems significant than those of the solution (Fig. R1a). However, the corresponding difference is actually just 3 ± 2 meV and can be explained by the small fluctuations in k_r and k_{nr} for the solution and the film.

Fig. R1 a, b, Transient PL decays of HzTFEX₂ in a deaerated toluene solution (a) and in a film with a PPF host matrix (b) at varying temperatures. **c,** Temperature-dependence of the time constants of delayed fluorescence (τ_{DF}) in a deaerated toluene solution (a) and in a film with a PPF host matrix. The solid lines in (c) represent the fits of τ_{DF} to a single exponential in inverse temperature.

If the answer to the above points is satisfactory, it is essential to refer to the film results in the main paper between the solution and device results. This could be comments on data in SI, but it's important to state ΔE_{ST} for the film and link it to the supplementary data.

We have thankfully included the below brief comments of the film data in the main paper (Page 5, line 159 in the manuscript). We have also moved the corresponding figures in SI to the Extended Data section in the main paper for their greater visibility to the reader.

=====

The negative ΔE_{ST} is retained in a solid-state host matrix (Extended Data Fig. 4 and Supplementary Information for details).

Extended Data Fig. 4 | Photophysical properties of HzTFEX₂ in a solid-state host matrix. **a**, Steady-state PL spectra of a thin film of bis(diphenylphosphoryl)dibenzo[*b,d*]furan (PPF):1wt% HzTFEX₂. **b**, Transient PL decays of PPF:1wt% HzTFEX₂ at varying temperatures under a N₂ atmosphere. **c**, Temperature-dependence of t_{DF} of PPF:1wt% HzTFEX₂. **d**, Temperature-dependence of k_{ISC} and k_{RISC} of PPF:1wt% HzTFEX₂. The solid lines in (d) represent the fits of the plots to the Arrhenius equation.

=====

Reviewer Reports on the Fourth Revision:

Referee #5

(Remarks to the Author)

I have reviewed the authors explanations and responses to the points raised, together with the revised manuscript. I find the adjustments sufficient to proceed to publication of these interesting and important results.